# Satellite Observations of Smoke-Cloud-Radiation Interactions Over the Amazon Rainforest

**Ross Herbert[1] and Philip Stier[1]**

[1] Atmospheric, Oceanic, and Planetary Physics, Department of Physics, University of Oxford, Oxford, OX1 3PU, United Kingdom

*Correspondence to*: Ross Herbert (ross.herbert@physics.ox.ac.uk)

## Abstract

The Amazon rainforest routinely experiences intense and long-lived biomass burning events that result in smoke plumes that cover vast regions. The spatial and temporal extent of the plumes, and the complex pathways through which they interact with the atmosphere, has proved challenging to measure and gain a representative understanding of smoke impacts on the Amazonian atmosphere. In this study we use multiple collocated satellite sensors onboard AQUA and TERRA platforms to study the underlying smoke-cloud-radiation interactions during the diurnal cycle. An 18-year timeseries for both morning and afternoon overpasses is constructed providing collocated measurements of aerosol optical depth (column integrated aerosol extinction, AOD), cloud properties, top-of-atmosphere radiative fluxes, precipitation, and column water-vapour content from independent sources.

The long-term timeseries reduces the impact of interannual variability and provides robust evidence that smoke significantly modifies the Amazonian atmosphere. Low loadings of smoke (AOD ≤ 0.4) enhance convective activity, cloudiness and precipitation, but higher loadings (AOD > 0.4) strongly suppress afternoon convection and promote low-level cloud occurrence. Accumulated precipitation increases with convective activity but remains elevated under high smoke loadings suggesting fewer but more intense convective cells. Contrasting morning and afternoon cloud responses to smoke are observed, in-line with recent simulations. Observations of top-of-atmosphere radiative fluxes support the findings, and show that the response of low-level cloud properties and cirrus coverage to smoke results in a pronounced and consistent increase in top-of-atmosphere outgoing radiation (cooling) of up to 50 Wm$^{-2}$ for an AOD perturbation of +1.0.

The results demonstrate that smoke strongly modifies the atmosphere over the Amazon via widespread changes to the cloud-field properties. Rapid adjustments work alongside instantaneous radiative effects to drive a stronger cooling effect from smoke than previously thought, whilst contrasting morning / afternoon responses of liquid and ice water paths highlight a potential method for constraining aerosol impacts on climate. Increased drought susceptibility, land-use change, and deforestation will have important and widespread impacts to the region over the coming decades. Based on this analysis, we anticipate further increases in anthropogenic fire activity to be associated with an overall reduction in regional precipitation, and a negative forcing (cooling) on the Earth's energy budget.

## 1. Introduction

Anthropogenic aerosols and their role in the earth system remain a key uncertainty in quantifying the impact of historic and future anthropogenic activity on the global climate (Forster et al., 2021). Aerosols interact with the atmosphere via modifying fluxes of solar and terrestrial radiation (referred to as

aerosol-radiation interactions, ARI) and by influencing the properties of clouds (referred to as aerosol-
cloud interactions, ACI), and therefore have the potential to significantly alter surface fluxes, cloud
properties, precipitation, and the energy budget of the atmosphere.
Biomass burning produces smoke aerosol particles that efficiently absorb shortwave radiation and
strongly perturb the atmosphere via both ARI and ACI processes. Smoke instantaneously reduces
shortwave radiation reaching the surface and produces localised warming of the smoke layer via the
ARI pathway. Rapid adjustments of the environment due to ARI can result in reduced surface fluxes
and supressed convection (Zhang et al., 2008b; Liu et al., 2020; Martins et al., 2009), with the localised
warming driving cloud evaporation or deepening depending on the cloud type and relative altitude of
the smoke (Koch and Del Genio, 2010; Herbert et al., 2020). Via the ACI pathway aerosol particles can
act as cloud condensation nuclei (CCN) or ice nuclei (IN) and instantaneously modify the number
concentration of cloud droplets or ice particles in a given cloud thus changing the cloud albedo. Rapid
adjustments associated with ACI include changes to precipitation efficiency and cloud evolution (Wu
et al., 2011; Liu et al., 2020; Thornhill et al., 2018; Marinescu et al., 2021; Zaveri et al., 2022). The
influence that a smoke particle has on a cloud and its environment is dependent on its physiochemical
properties, which determine its optical properties and ability to act as a CCN or IN. These properties
are dependent on the type of fuel (McClure et al., 2020; Petters et al., 2009), the combustion efficiency
(Liu et al., 2014), and may also change with time through aging processes and interaction with other
species (Vakkari et al., 2014; Zhang et al., 2008a). This, combined with the myriad of pathways through
which smoke can impact the environment, and the spatial and temporal extent of the smoke plumes, has
proven a challenge to understand at a process level and represent in atmospheric models. As a result,
there remains considerable uncertainty in our understanding of smoke impacts to climate on a global
scale (Forster et al., 2021; Bond et al., 2013), which will become increasingly important in the future
as drought conditions become more prevalent (Stocker et al., 2013) and anthropogenic deforestation
continues (de Oliveira et al., 2020).
The Amazon rainforest in South America is one of the world's largest sources of biomass burning
aerosol (van der Werf et al., 2017), with peak emissions observed during the annual dry season (August
to October) driven almost exclusively by agricultural activities and anthropogenic activity (Libonati et
al., 2021). The associated smoke plumes can extend high into the troposphere (Holanda et al., 2020)
and cover vast regions, with sustained high atmospheric loadings of smoke often observed for days to
weeks. Observational studies have demonstrated the ability for smoke to strongly influence the Amazon
atmosphere during the dry season via changes to the initiation and efficiency of precipitation processes
in deep convective clouds (Andreae et al., 2004; Gonçalves et al., 2015; Camponogara et al., 2014;
Bevan et al., 2008; Braga et al., 2017; Wendisch et al., 2016). These impacts are largely attributed to
the suppression of convection or enhanced cloud droplet number concentrations, though the overall
response of cumulative precipitation remains uncertain.
The widespread and long-lived nature of the smoke perturbations present a challenge to make the
necessary in-situ measurements that capture the overall impact of the smoke on the atmosphere.
Regional modelling studies with sufficient complexity to reproduce the convective nature of the
Amazon atmosphere have been used to quantify the widespread smoke-cloud-radiation interactions. A
consistent result is widespread suppression of convection underneath smoke plumes due to the cooler
surface and elevated heating stabilising the boundary layer, and a corresponding reduction in cumulative
precipitation (Martins et al., 2009; Zhang et al., 2009; Wu et al., 2011; Liu et al., 2020; Herbert et al.,
2021). There is less agreement on the change to the widespread cloud field properties such as cloud
fraction (CF), liquid water path (LWP), and ice water path (IWP), potentially due to the complexity of
sufficiently representing ACI and ARI processes in these models (Marinescu et al., 2021; White et al.,
2017). In a recent study Herbert et al. (2021) performed week-long simulations of smoke-cloud-
radiation interactions over the Amazon at convection-permitting resolution. The authors reported
considerable diurnal variation in the cloud response with enhanced cloudiness overnight and reduced
cloudiness in the afternoon; this occurred alongside a gradual increase in the IWP across the domain
that strongly dictated the overall positive effective radiative forcing (ERF) due to the smoke. The
response in IWP was in contrast to a similar study by Liu et al. (2020) who reported only weakly
increasing IWP across the model domain, with changes in the liquid cloud fraction dictating the overall
negative ERF. The contrasting results have important implications for the ERF of smoke, however,
without robust observational information it is difficult to establish whether these model-based
conclusions are valid.
One means of gathering this information is using space-borne remote observations that are able to
provide widespread and routine coverage. Koren et al. (2004) used retrievals from the Moderate
Resolution Imaging Spectroradiometer (MODIS) instrument onboard the AQUA satellite to examine
cloud-smoke relationships during the 2002 dry season; the authors found that the low-cloud fraction
was strongly supressed as the smoke optical depth increased. Yu et al. (2007) similarly used MODIS-
AQUA retrievals to examine widespread smoke-cloud interactions for the 2002 and 2003 dry seasons
and found pronounced variability in the smoke-cloud relationships between the years studied, and
considerable sensitivity to the cloud properties (e.g., LWP) in both years. This study supported the
results from Koren et al. (2004), but also demonstrated important interannual variability, suggesting a
longer timeseries is required to quantify and understand the underlying processes through which the
smoke perturbs the widespread environment. Koren et al. (2008) used MODIS-AQUA retrievals of
cloud fraction and cloud top height during the dry seasons of 2005 to 2007 to propose that the response
of clouds to smoke is nonlinear: at low loadings of smoke clouds are invigorated, but at higher loadings
the clouds are suppressed. These results were supported by a simplified theoretical model that
additionally suggested the invigoration was driven by ACI processes, whereas the suppression was
driven by ARI processes. These widespread remote observations provide valuable insight but there are
several areas that can be improved upon: 1) Interannual variability – the response of the atmosphere to
smoke may be different from one year to the next, which may mask the underlying smoke-cloud-
radiation processes and overall impact of the smoke. 2) Diurnal cycle – modelling studies suggest
important diurnal responses to the cloud and precipitation (Liu et al., 2020; Herbert et al., 2021), yet
previous remote observations over the Amazon have only observed a very small window of time
coinciding with the AQUA satellite overpass time (~1330 local solar time; LST). 3) Radiative effect –
it is understood that smoke may have important impacts to deep convective clouds and their optical
properties, yet previous studies have estimated radiative effects using offline radiative transfer models,
which may not be representative of the true radiative effect
In this study we build upon previous efforts to quantitively understand aerosol-cloud-radiative
interactions by focusing on smoke impacts to the Amazonian atmosphere during the dry season. This
region provides a unique opportunity to study the interactions between long-lived, substantial aerosol
loadings and deep convective clouds over a widespread region. We use 18 years of satellite observations
to produce a 1-degree gridded climatology of smoke-cloud-radiation effects over the Amazon during
the biomass burning season. The long timeseries allows us to work towards removing or reducing the
interannual variability, and provides the means to robustly explore more of the parameter space. We
explore the diurnal cycle of the responses to smoke by combining and contrasting the AQUA satellite
retrievals with the TERRA satellite, which is host to the same instruments as AQUA but has an overpass
time of ~1030 LST. The two satellites are host to several instruments including MODIS, CERES
(Clouds and the Earth's Radiant Energy System), and AIRS (The Atmospheric Infrared Sounder). The
use of all three instruments, alongside reanalysis and precipitation datasets, provides spatially and
temporally collocated data that can be used to support individual observations and strengthen the
analysis. Additionally, CERES can provide collocated information on the top-of-atmosphere (TOA)
radiative fluxes and the overall radiative effect of the smoke, which has not previously been explored
in this region.

## 2. Methodology

## 2.1 Domain and Analysis Time Period

Biomass burning occurs annually during the dry season between the months of August and October (**Figure 1**d). We focus our analysis on the peak AOD month of September between 2002 and 2019, and confine the analysis to an area (70W to 52W, 15S to 1S) collocated with a region of climatologically high AOD **Figure 1**a). AERONET stations at the Rio Branco and Alta Floresta sites provide information on the single scattering albedo (SSA) of the aerosol throughout the analysis period. These sites are situated at opposite ends of the analysis region and collocated with the climatologically highest regions of AOD (**Figure 1**a). Histograms of the daily-mean SSA from each station, given at 675 nm, are shown in **Figure 1**e. Both stations show $SSA_{675}$ ranging from values as low as 0.85 to 0.98, with a peak around 0.93. This consistent with in-situ local observations of smoke optical properties (Palácios et al., 2020; Rosário et al., 2011), providing good evidence that the aerosol in this analysis period and domain is strongly absorbing smoke. Note that mineral dust has a SSA closer to 1 at this wavelength (Di Biagio et al., 2019). We would therefore expect ARI mediated impacts via absorption of solar radiation to be a viable mechanism in this region.

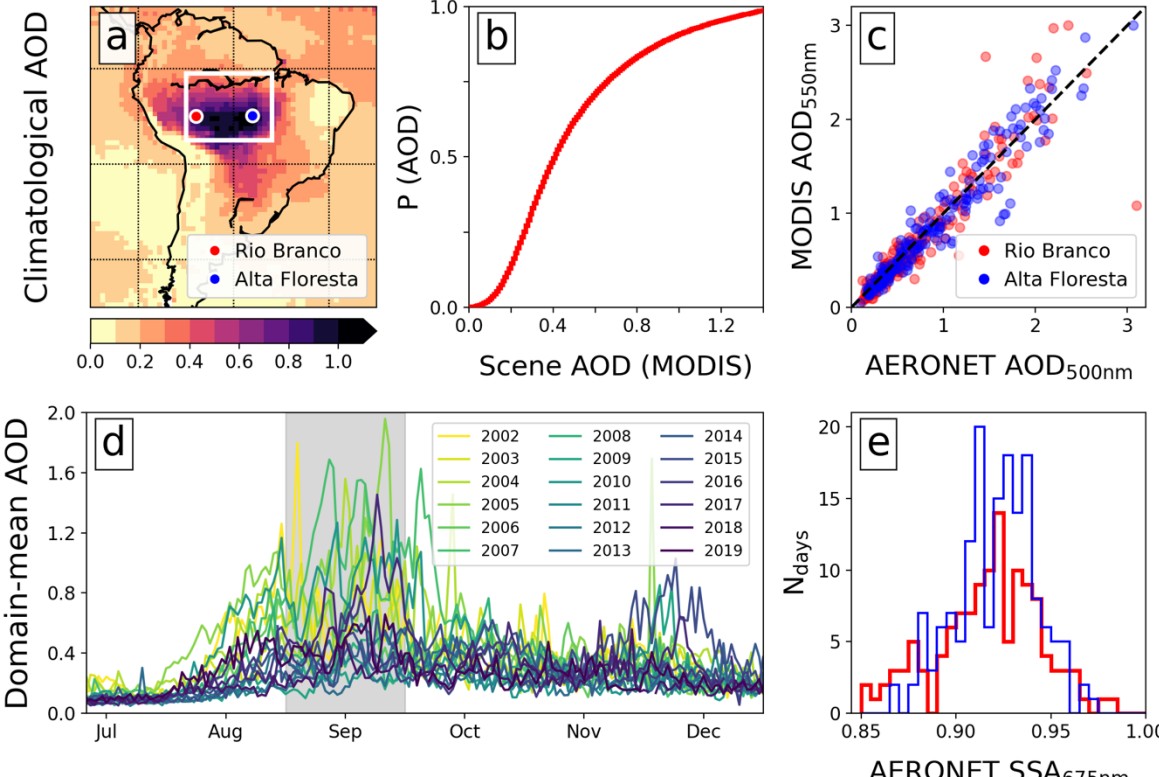

**Figure 1**. Information on the climatological AOD and SSA in the region during the analysis period of 2002 to 2019: (**a**) MODIS AOD climatology for September; (**b**) cumulative probability of occurrence of gridded MODIS AOD in the analysis domain (white box in a); (**c**) collocated AERONET and daily-mean MODIS retrieved AOD at the two stations shown in a; (**d**) timeseries of daily-mean AOD from MODIS-AQUA over the analysis region (timeseries only shown between July and December for clarity), and (**e**) histograms of the daily-mean SSA at 675nm from the two AERONET stations. MODIS AODs are given at a wavelength of 550nm and AERONET at 500nm.

## 2.2 Satellite and Reanalysis Products

In this study we primarily use data products from the MODIS, CERES, and AIRS instruments onboard AQUA and TERRA satellites. This is complemented by precipitation information from the Global Precipitation Measurement (GPM) level 3 Integrated Multi-satellitE Retrievals for GPM (IMERG) dataset, and meteorological information from ERA5 reanalysis. A brief overview of the variables extracted from each dataset is presented below, with full details in Table S1.

*MODIS AQUA and TERRA*: We use the MODIS collection 6.1 1-degree level 3 products (AQUA: MYD08_D3 and TERRA: MOD08_D3) for instantaneous retrievals of AOD (given at 550 nm) and cloud properties including total cloud fraction ($CF_{total}$), liquid cloud fraction ($CF_{liquid}$), LWP, IWP, total water path (TWP), cloud top temperature (CTT) and height (CTH), cloud optical thickness of both liquid and ice ($COT_{total}$) and liquid only ($COT_{liquid}$), ice cloud droplet effective radius ($RE_{ice}$), and cirrus fraction ($CF_{cirrus}$). The morning TERRA overpass is at ~1030 LST and afternoon AQUA overpass at ~1330 LST. We also use the level 2 products MYD04_L02, MOD04_L02, MYD06_L02, and MOD06_L02 to obtain aerosol and cloud properties at a finer resolution (10 km) for comparison with the coarser scale level 3 dataset.

*CERES top-of-atmosphere fluxes*: Top of atmosphere fluxes of radiation for the incoming solar ($SOL_{TOA}$), shortwave ($SW_{TOA}$), longwave ($LW_{TOA}$), and net ($NET_{TOA}$) components on a 1-degree grid are taken from the CERES level 3 data product, SSF1Deg-1H, that provides instantaneous fluxes onboard AQUA and TERRA satellites.

*AIRS*: The AIRS daily level 3 product, AIRS3STD, is used to provide daily mean values of total column water vapor ($QV_{column}$), surface level specific humidity ($QV_{surface}$), and surface level relative humidity ($RH_{surface}$).

*IMERG precipitation*: Daily accumulated precipitation estimates ($P_{accum}$) on a 0.1-degree grid are taken from the IMERG dataset (3B-DAY_MS_MRG_3IMERG_V06). A second dataset (3B-HHR_MS_MRG_3IMERG_V06B) provides 30-minute temporal resolution estimates at 0.1-degree resolution, used to determine cumulative precipitation in the morning ($P_{AM}$; 0700 – 1200 LST), afternoon ($P_{PM}$; 1400 – 1900 LST), and peak precipitation rate during the diurnal cycle ($P_{peak}$).

*ERA5 Reanalysis:* Daily mean 850 hPa horizontal winds and 2m temperature ($T_{2m}$) on a 1-degree grid are taken from the ERA5 reanalysis dataset for spatial collocation with satellite observations; horizontal wind components are used to determine the wind direction (degrees from due north). Daily mean fields of 850 hPa specific humidity ($QV_{850}$) and temperature ($T_{850}$) are also taken from the dataset to obtain large-scale environmental conditions upstream of the domain, discussed in Section 3.6; mean values are determined over a region off the east coast of South America (35W to 30W, 25S to 5N), roughly five days upstream of the prevailing winds (see supplementary material).

## 2.3 Collocating Datasets

All data is analysed on a regular 1-degree grid. MODIS, CERES, AIRS, and ERA5 datasets are provided on a 1-degree grid so are readily collocated spatially, and IMERG data is regridded onto a 1-degree grid. CERES instantaneous TOA fluxes and MODIS products each have separate datasets for TERRA and AQUA overpasses and are temporally collocated in the analysis. Daily mean ERA5 horizontal winds, describing the large-scale daily-mean flow, are selected for each corresponding day of the timeseries, and daily mean AIRS and IMERG daily $P_{accum}$ and $P_{peak}$ data similarly selected. $P_{AM}$ and $P_{PM}$ are collocated with the TERRA and AQUA datasets, respectively. Although AIRS provides instantaneous retrievals we use the retrieved atmospheric water variables to describe the large-scale environmental properties, and as such do not require the higher temporal resolution.

For this study we are primarily interested in how widespread properties of the atmosphere change
with AOD. We use AOD as a proxy for the availability of aerosols that can influence clouds both via
ARI and ACI, thereby assuming that as AOD increases linearly, so does the number of aerosols that act
as CCN and interact with radiation. For ARI this assumption is reasonable if the source and size
distribution stays relatively constant as AOD increases. As the primary source of aerosol in this region
is biomass burning, with AOD increasing linearly with the frequency of fires (Ten Hoeve et al., 2012),
this is to first-order a reasonable approximation. This can be similarly applied to the availability of CCN
but the number activated is also dependent on properties of the atmosphere, namely the updraught speed.
Herbert et al. (2021) used in-situ observations from field campaigns over the Amazon and found a
positive albeit non-linear relationship between AOD and cloud droplet number concentration (CDNC).
However, this is confounded by any changes to the distribution of vertical velocities as AOD changes.
Given the inherent non-linearity and confounding factors between AOD and CDNC we can only say
that AOD is a reasonable proxy for the availability of CCN.
In this analysis representation error may arise from the fact that AOD retrievals are made in clear-
sky conditions, whereas cloud properties are necessarily in cloudy sky. Wet scavenging is known to
impact the column loading of aerosol (Gryspeerdt et al., 2015), therefore can we be confident that the
AOD retrievals are representative of the underlying conditions impacting the clouds? As precipitation
predominantly occurs within the afternoon period a comparison of AOD retrieved in the TERRA and
AQUA overpasses provides some information as to whether we may expect wet scavenging to be
strongly influencing the AOD. Figure S1 in the supplementary material shows that there is very little
systematic bias between the two overpasses, even though precipitation has likely occurred in some of
the scenes, therefore giving us confidence that clear-sky retrievals of AOD are representative of the
widespread AOD. A second source of potential bias may arise from the retrieval of AOD in cloudy
conditions. The presence of aerosols in the vicinity of clouds can impact the retrieval of both properties:
enhanced humidity close to clouds can cause aerosols to swell elevating the AOD retrievals, whilst
aerosols embedded within, or below, clouds may be misidentified as cloud, thereby modifying the
retrieved cloud optical properties. Finally, very high loadings of aerosols may be misidentified as cloud.
These are well-known sources of retrieval bias and as such cloud masking algorithms are continually
refined to separate the influence of the two. The MODIS cloud mask product in the collection 6 variants,
used in this study, is constructed using 1km scale pixels and employs multi-spectral tests to identify
heavy aerosol loading. Aerosol retrievals are made in clear-sky pixels, with collection 6.1 using the
Dark-Target and Deep-Blue aerosol retrieval algorithms, designed to take into account the underlying
surface properties. These well maintained, and extensively evaluated products (e.g., Wei et al., 2019;
Huang et al., 2019; Zhang et al., 2022; Levy et al., 2013; Platnick et al., 2017) provide a robust dataset
of collocated aerosol and cloud properties but may not remove all bias. Therefore, to support our
analysis we will pay particular attention to aerosol-cloud misclassification, especially at high cloud
fractions. We achieve this by first comparing the MODIS retrievals of AOD with those from two
AERONET stations (below), and later in Section 4 repeat the analysis with level 2 data products, where
we find the same conclusions.
In previous studies (e.g., Koren et al., 2004; Yu et al., 2007) scenes where cloud fraction exceeds 0.8
have been removed to avoid AOD retrieval uncertainty, yet in this study we do not in order to preserve
the data and avoid potential bias to the properties of the cloud field. This ensures that we are considering
the response of the atmosphere over the region as a whole, rather than a subset. If clouds were strongly
influencing the retrieved AOD then independent retrievals from AERONET, able to take measurements
throughout the day, would highlight biases. A spatial and temporal collocated comparison of AOD
retrieved from two AERONET stations (Rio Branco and Alta Floresta) with mean MODIS AOD shown
in **Figure 1**d gives confidence that MODIS AOD retrievals are not biased high in the presence of high
cloud coverage. This is consistent with the low biases reported by (Wei et al., 2019; Sayer et al., 2019),
who additionally show evidence that South America has one of the lowest regional biases between the
two datasets, partly due to the performance of the MODIS AOD retrievals over forested land.
The vertical profile of aerosol is a difficult property to measure on the scales that we are interested
in, yet previous studies (e.g., Koch and Del Genio, 2010) have shown that the position of smoke in
relation to clouds can greatly impact the cloud rapid adjustments and ERF. Most significantly, when
smoke is elevated above clouds it reduces the scene albedo, thereby driving a positive TOA
instantaneous radiative effect. Gonzalez-Alonso et al. (2019) used three remote sensing instruments
over 6 years to construct a climatology of smoke heights over the Amazon. The authors found that
smoke plumes during September are generally located below 1.5 km, with less than 5 % of smoke plume
injection heights observed in the free troposphere. Some studies, focusing on the eastern edge of the
Amazon rainforest, have reported the presence of smoke being transported from the African continent
at concentrations that often compete with localised sources (Barkley et al., 2019; Holanda et al., 2020).
Therefore, although we assume that the smoke in this analysis is predominantly within the BL and from
local sources, we caveat that this is not always the case. We discuss the validity of this assumption in
Section 3.5.
## 3. Results

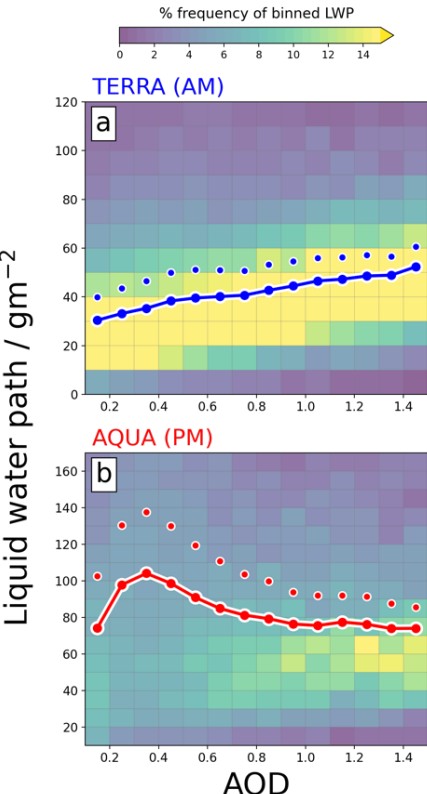

**Figure 2**. MODIS liquid water path as a function of AOD for the (**a**) morning TERRA overpass and (**b**) afternoon
AQUA overpass. Joint-histograms show % frequency of LWP binned by AOD, and coloured lines (individual
circles) show the geometric (arithmetic) mean in each AOD bin. Data is only shown for cloudy scenes where
LWP > 0.
## 3.1 Liquid Water Path
The 18-year September climatology shows that there are contrasting LWP-AOD relationships in the
morning and afternoon. **Figure 2** shows a consistent increase in LWP with AOD for the morning
overpass (panel a); the histogram suggests that the existing clouds become increasingly laden with water
as AOD increases. Conversely, the afternoon overpass (panel b) shows an initial spread of the cloud
distribution to higher LWP, followed by a gradual focus towards lower LWP. This behaviour describes
an initial enhancement followed by gradual suppression. The same analysis, performed on the domain-
mean dataset rather than the 1-degree grid, results in the same relationships (see Figure S2).

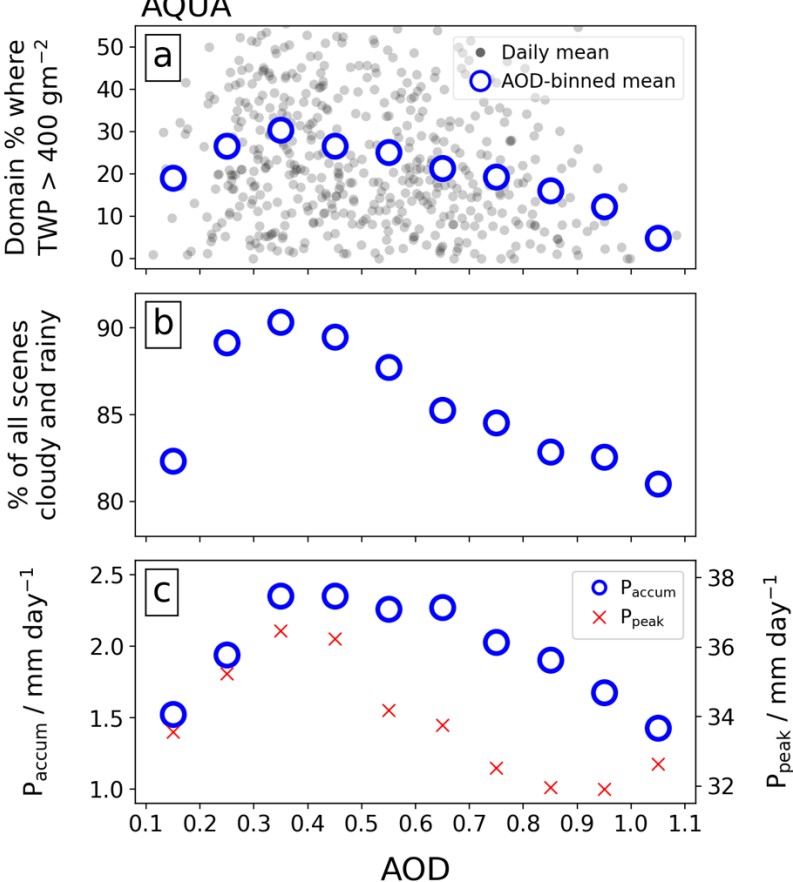

**Figure 3**. Convection and precipitation as a function of AOD: (**a**) percentage of the domain where TWP >
400 gm$^{-2}$ as a function of the domain-mean AOD for each day (filled grey circles) and the mean of all days
binned by AOD (empty blue circles); (**b**) mean percentage of all scenes that include liquid cloud and
precipitation as a function of binned AOD; and (**c**) mean daily accumulated precipitation (blue circles) and daily
peak precipitation (red crosses) in each scene binned by AOD. MODIS data is shown for the AQUA overpass.
310        The contrasting diurnal responses of LWP to AOD are consistent with the high-resolution modelling
study from Herbert et al. (2021). In their study it was found that the domain-mean LWP adjustment to
an AOD perturbation was positive in the morning (due to widespread modification to the
thermodynamic environment) but negative in the afternoon (due to a suppression of convection).
315        The enhanced mean LWP in the morning overpass (**Figure 2**a) is consistent with ACI-induced
suppression of the warm-rain process, where an increase in CCN from smoke results in a more
numerous, smaller, cloud droplets; this behaviour has been observed in observational (e.g., Twohy et
al., 2021; Andreae et al., 2004; Martins and Silva Dias, 2009) and modelling (e.g., Liu et al., 2020;
Herbert et al., 2021; Martins et al., 2009) studies. The afternoon AOD dependence in **Figure 2**b is well
aligned with changes in the convective activity. An increase in CCN availability has been found to
promote convection in some studies via ACI adjustments (Fan et al., 2018; Lebo, 2018; Khain et al.,
2005; Marinescu et al., 2021), whilst the heat generated from biomass burning has also been found to
enhance buoyancy and deep convection (Zhang et al., 2019). ARI adjustments from smoke also impact
convection as the aerosol particles cool the surface and stabilise the boundary layer via elevated heating
of the absorbing aerosol, acting to suppress convection. Using a theoretical model Koren et al. (2004)

demonstrated that the competition between ACI and ARI adjustments in deep convective clouds results in an initial enhancement (driven by ACI) for small AOD perturbations, followed by a suppression at higher AOD as ARI adjustments dominate. The observations in this study are consistent with this; **Figure 3**a demonstrates that the percentage of the domain that exhibits high TWP loadings (indicative of deep convective clouds) follows this non-linear relationship with AOD. **Figure 3**b and **Figure 3**c additionally show that the non-linearity is reflected in the occurrence of precipitating liquid clouds, and the magnitude of precipitation itself ($P_{accum}$ and $P_{peak}$). For AOD > 0.4 there is less suppression in the precipitation (**Figure 3**b and c) compared to the fraction of domain that shows signs of convective activity (**Figure 3**a); this may suggest that at high AOD there are fewer deep convective cells but those that do form are more intense, providing relatively more precipitation per convective cell.

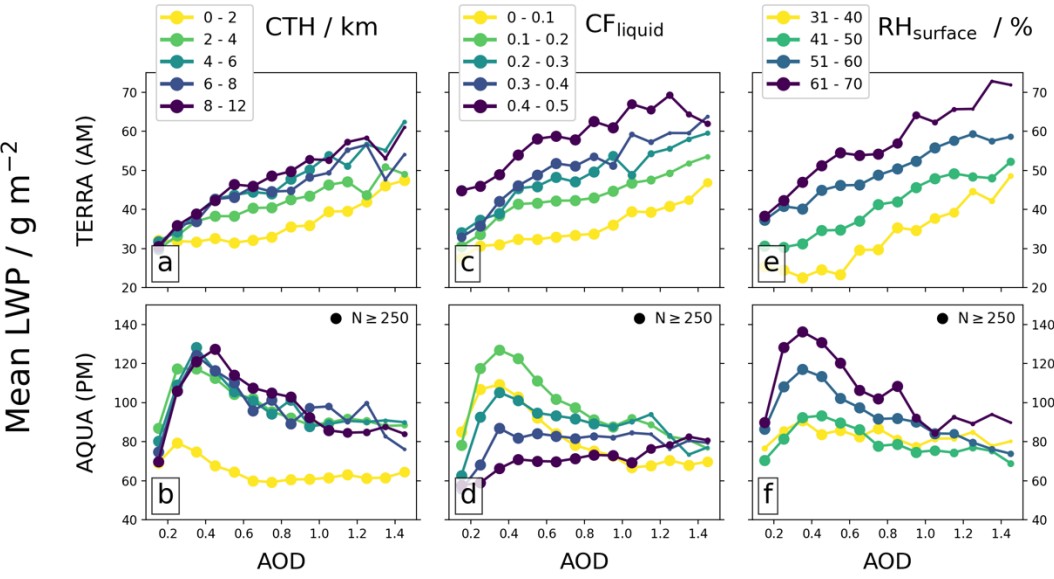

**Figure 4**. Geometric mean LWP as a function of AOD, subset by different cloud or environmental properties: MODIS cloud top height (**a – b**), MODIS liquid cloud fraction (**c – d**); and AIRS surface level RH (**e – f**). For each plot the top panel is for the TERRA overpass and bottom panel for the AQUA overpass. The size of each circle gives a representation of how many scenes are included in the mean, with a maximum size shown for $N \geq 250$.

Subsetting the dataset by CTH, $CF_{liquid}$, and $RH_{surface}$ in **Figure 4** demonstrates that the LWP-AOD relationships observed in **Figure 2** persist when constrained by environmental conditions. For the morning TERRA overpass (**Figure 4** top row) the AOD-binned mean LWP increases with AOD for all constrained datasets. CTH (**Figure 4**a) and $CF_{liquid}$ (**Figure 4**b) show interesting behaviour. For AOD < 0.4 LWP increases sharply for all clouds that extend beyond 2 km and exhibit $CF_{liquid}$ > 0.2, which may indicate mesoscale systems that have persisted overnight. Above this AOD (where we posit daytime convection is being suppressed) the data is predominantly confined to small boundary layer clouds with $CF_{liquid}$ < 0.1 and CTH < 1 km (the smaller marker sizes depict fewer data points), suggesting a link between convective activity during the daytime (**Figure 3**) and the development of larger mesoscale systems. Subsetting by $RH_{surface}$ (**Figure 4**c) shows a consistent and positive LWP-AOD relationship, which suggests the increase in LWP is being driven by changes in cloud properties (ACI), rather than the environment. The AQUA overpass in the afternoon (**Figure 4** bottom row) provides more evidence that the mean response is controlled by the initial enhancement (AOD < 0.4) and then suppression (AOD > 0.4) of convective activity. First, all clouds that exceed CTH of 2 km display an almost identical relationship with LWP (**Figure 4**b), with this subset of clouds typically representative of locations containing cells of deep convection. Second, lower $CF_{liquid}$ scenes (**Figure**

**4**d) show greater sensitivity to AOD and greater magnitudes of LWP. This can be explained by appreciating that deeper convective clouds will contain more cloud condensate in the ice phase, and therefore not retrieved as liquid cloud (subsetting IWP by $CF_{total}$ confirms this) – low $CF_{liquid}$ scenes with high loadings of LWP thereby indicate regions with intense convective cells. Subsetting by $RH_{surface}$ demonstrates that the environmental conditions play a role in the LWP-AOD relationship, and is likely mediated by the connection between boundary layer moisture, CAPE, and convective activity (a similar relationship was observed by Ten Hoeve et al. (2011) for $COT_{liquid}$). The response of $RH_{surface}$ to AOD will be discussed in Section 3.4.

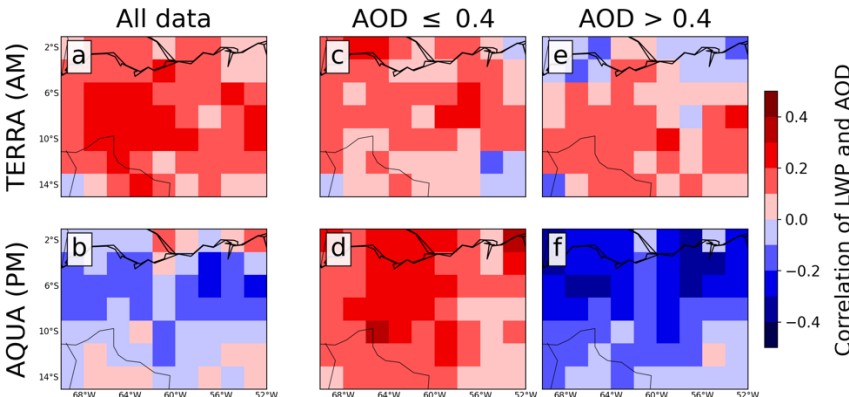

**Figure 5**. Pearson's correlation coefficient between LWP and AOD. Top row (**a**, **c**, **e**) shows the TERRA overpass in the morning, and bottom row (**b**, **d**, **f**) shows the AQUA overpass in the afternoon. Left column (**a** and **b**) shows the spatial distribution of the coefficient for all data, middle column (**c** and **d**) shows data for AOD ≤ 0.4, and right column (**e** and **f**) shows data for AOD > 0.4. Red colours depict a positive correlation, blue colours a negative correlation.

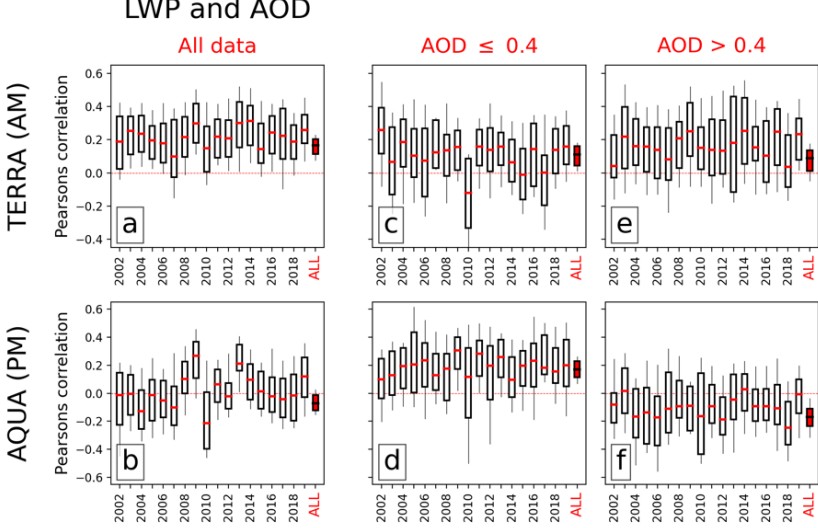

**Figure 6**. Boxplots showing Pearson's correlation coefficients in the domain, for the September of each individual year during the timeseries. Rows and columns as in **Figure 5**. Right-most boxplot (in red) in each subplot shows the data for all years.

Is this response spatially consistent? If not it may suggest we are seeing different regions of the domain influencing the mean and masking any underlying AOD relationship. **Figure 5** shows the Pearson's correlation coefficient between LWP and AOD across the domain (regridded from 1-degree to 2-degree resolution to increase the number of datapoints). The TERRA correlation apparent in **Figure**

**2**a suggests a consistent positive relationship throughout the range of AOD, which is also observed
across the domain in **Figure 5**a with positive (albeit small) correlation coefficients throughout. For the
AQUA overpass we also see a consistent correlation as observed in **Figure 2**: for AOD ≤ 0.4 the
correlation is consistently positive across the domain (**Figure 5**d), and AOD > 0.4 is consistently
negative throughout the domain (**Figure 5**f).
The interannual variability in the LWP-AOD relationship during September is shown in **Figure 6**
(also see Figure S3). Here the correlation coefficients are similarly determined throughout the domain
(as in **Figure 5**) but now data is additionally subset for each year. The TERRA overpass shows a positive
LWP-AOD relationship (for all AOD) over the entire timeseries (**Figure 6**a) with some degree of
interannual variability. Note the final boxplot using the entire 18-year timeseries, demonstrating the
benefit of using a long timeseries. The AQUA afternoon overpass shows more interannual variability,
though still shows a consistent relationship below (**Figure 6**d) and above AOD = 0.4 (**Figure 6**f). A
possible explanation for the additional variability is if the LWP response is connected to the
enhancement or suppression of convective cells; the CAPE and other environmental conditions required
for triggering deep convection would be sensitive to larger-sale drivers and thus influence the number
of convective cells on any given day, month, and year. As shown in **Figure 4**f the LWP response in the
afternoon is particularly sensitive to the $RH_{sfc}$ (moisture content is a key component of CAPE). In
Section 3.6 we will provide evidence that suggests the LWP-AOD relationships presented here are not
driven by large-scale external drivers, and are primarily an internalised response to AOD.

## 3.2 Ice Water Path and Effective Radius

The simulations from Herbert et al. (2021) showed pronounced increases in IWP and ice-cloud
coverage which had important implications for the longwave TOA radiative effect due to smoke. **Figure**
**7** shows the mean IWP and $RE_{ice}$ retrieved from MODIS binned by AOD. For the IWP in the morning
overpass (**Figure 7a**) there is an overall positive relationship with > 50% increase in IWP from
AOD = 0.2 to AOD = 1.0. The AQUA overpass (**Figure 7b**) shows initial enhancement of IWP up to
AOD = 0.4 followed by a consistent negative relationship. In both timeframes, the geometric mean
displays a maximum IWP response to AOD of +50%, though there is considerably more sensitivity to
AOD in the afternoon. This behaviour is closely correlated with the LWP-AOD relationships (**Figure**
**2**). Enhanced LWP in the morning from a suppressed warm-rain process allows more condensate to
reach the freezing level, and in the afternoon changes in convective activity will have a direct influence
on the amount of condensate reaching the freezing level.
$RE_{ice}$ provides information on the cloud-top ice particle size distribution; **Figure 7**c and **Figure 7**d
show mean $RE_{ice}$ as a function of AOD for bins of IWP. For AOD < 0.4 $RE_{ice}$ decreases with AOD for
all IWP bins during both overpasses, whilst at AOD > 0.4 $RE_{ice}$ increases for low IWP scenes, and
continues to decrease for high IWP scenes. This behaviour suggests that for deep convective clouds
associated with high IWP, increasing AOD and the availability of CCN results in smaller ice particle
sizes at the cloud-top. A possible explanation is ACI effects resulting in a larger CDNC of smaller
droplet sizes at the freezing level; smaller ice particles increase the longevity of deep convective outflow
and high-altitude cloud coverage (Wendisch et al., 2016). Lower IWP scenes ($< 100$ g m$^{-2}$) generally
show an increasing $RE_{ice}$ with AOD; these scenes may be associated with weakly convective regions
dominated by shallower convection. This contrasting behaviour is consistent with Zhao et al. (2019)
who found ice particle size decreased for strongly convective regions, and increased for moderately
convective regions (when going from clean to polluted conditions), which occurred due to the different
freezing pathways dominant in each type of convection.

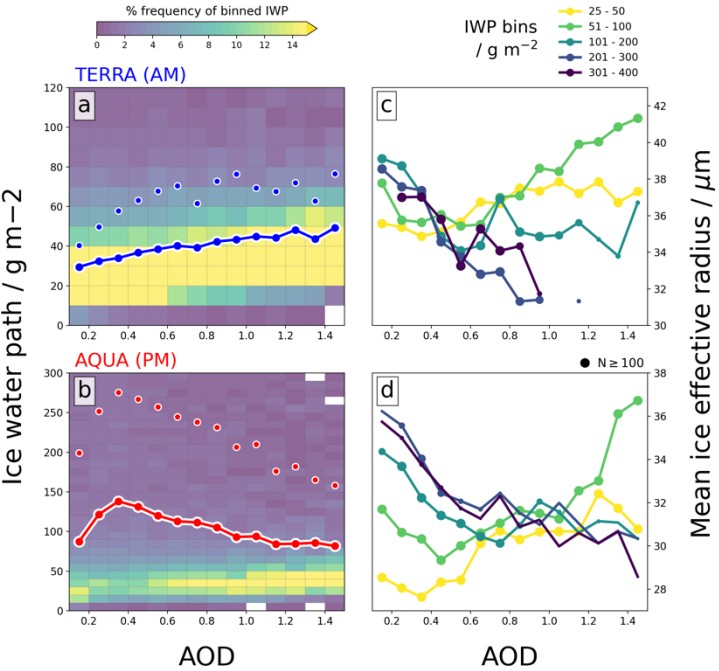

**Figure 7**. MODIS ice water path (IWP) as a function of AOD (**a** and **b**) and MODIS mean ice effective radius binned by IWP as a function of AOD (**c** and **d**) for the TERRA (top row) and AQUA overpasses (bottom row). Joint-histograms show % frequency of IWP binned by AOD, and coloured lines (individual circles) show the geometric (arithmetic) mean in each AOD bin.

## 3.3 Cloud Fraction

Changes to the cloud coverage over a region strongly influences the TOA radiative response. Subsetting $CF_{liquid}$ and $CF_{total}$ to low (0.0 < AOD < 0.2), mid (0.3 < AOD < 0.5), and high-AOD (0.8 < AOD < 1.0) scenes in **Figure 8** demonstrates widespread modifications to the cloud field over the region, alongside the changes in LWP.

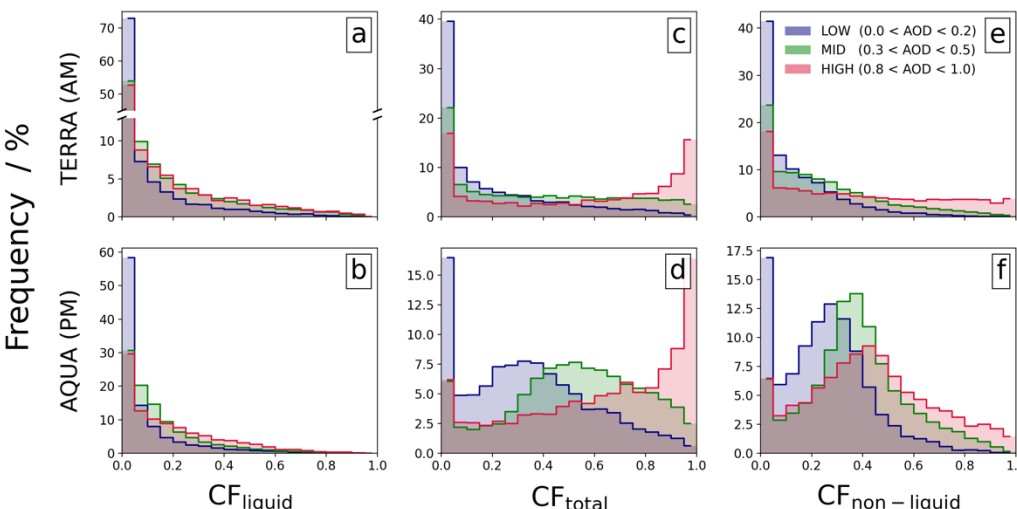

**Figure 8**. Normalized probability of occurrence of $CF_{liquid}$ (**a – b**), $CF_{total}$ (**c – d**) and $CF_{non-liquid}$ (**e – f**) for low (0.0 < AOD < 0.2), mid (0.3 < AOD < 0.5) and high (0.8 < AOD < 1.0) AOD scenes. Top row shows the TERRA overpass in the AM, and the bottom row shows the AQUA overpass in the PM. Note the break in the y-axis in panel (**a**).

The relative percentage of cloud-free scenes (CF < 0.05) in both overpasses and all cloud phases
strongly decrease for AOD > 0.2. $CF_{liquid}$ in the morning (**Figure 8**a) is well aligned with LWP (**Figure**
**2**a) and a suppression of precipitation, promoting cloudiness. There is little change going from mid to
high-AOD scenes suggesting a saturation effect of $CF_{liquid}$, though the cloud LWP (**Figure 2**a) continues
to increase; this could be associated with a widespread and robust drying of the boundary layer as AOD
increases (see Section 3.4). In the afternoon $CF_{liquid}$ is similarly well-correlated with the LWP-AOD
relationship and convective activity. For mid-AOD scenes the enhanced convection drives an increased
frequency of liquid cloud coverage over much of the distribution (**Figure 8**b). At higher AOD loadings
there is a suppression of convection which promotes the occurrence of liquid cloud retrievals
(fewer/weaker convective cells results in reduced mixed-phase cloud coverage). This result may have
important implications for the TOA radiative response as liquid clouds are more radiatively opaque than
ice clouds (Cesana and Storelvmo, 2017).
$CF_{total}$ (**Figure 8** c – d) demonstrates widespread sensitivity of the Amazon to the presence of AOD,
with clear shifts in the cloud field distribution that correlate with the changes in convective activity.
$CF_{non-liquid}$ ($CF_{total} - CF_{liquid}$) provides information on the non-liquid phase cloud coverage (**Figure 8** e –
f). Beginning with the afternoon overpass (**Figure 8**d), low AOD scenes are characterised by ~85%
cloud coverage, with a peak centered around $CF_{total} = 0.3$; this likely correlates with the presence of
scattered deep convective cells that extend beyond the freezing level. As convective activity increases
with AOD (**Figure 3**) the domain becomes cloudier, and the peak occurrence (for mid-AOD) shifts to
higher coverage as enhanced convection promotes more numerous / intense cells, increasing the cloud
coverage. For high-AOD scenes the convection is suppressed, promoting the occurrence of extensive
$CF_{total}$ coverage exceeding 0.8 (aerosol misclassification of cloud was discussed in Section 2.3, and we
do not believe it is heavily influencing the coincident high-AOD and high-$CF_{total}$ scenes). $CF_{non-liquid}$
(**Figure 8**f) shows that although convection is supressed in the high-AOD scenes (lower peak) there is
still some convective activity with a peak centered at higher cloud coverages, which suggests that under
high-AOD conditions deep convection is less likely, but when it does occur it is more intense. This may
explain why mean $P_{accum}$ remains relatively enhanced for AOD > 0.4 (**Figure 3**b and c) even though
convective activity is less likely (**Figure 3**a), though would require more attention to confirm. $CF_{non-}$
$_{liquid}$ demonstrates that the extensive $CF_{total}$ coverage under high-AOD conditions in **Figure 8**d is driven
by a combination of liquid and non-liquid clouds, and not solely due to extensive cirrus clouds or deep
convective anvil outflow. The morning overpass $CF_{total}$ and $CF_{non-liquid}$ (**Figure 8**c and e) bear strong
similarities to the afternoon overpass: at low AOD there are preferentially more low-coverage scenes,
and at high AOD there are preferentially more high-coverage scenes. The sensitivity to AOD is
primarily driven by the non-liquid phase and likely associated with the previous day's convective
activity; this is most evident under high-AOD scenes and may indicate longer-lived convective systems
or more intense cells. The pronounced shift from low to high $CF_{total}$ occurrence with AOD during both
overpasses (**Figure 8**c and d) is a consistent feature for all years when individually analysed (see Figure
S4), which suggests this is a causal relationship, rather than an artefact of co-varying meteorology which
is more likely to exhibit interannual variability. This will be explored further in Section 3.6.

## 3.4 Large-scale Environment

Changes to the large-scale environments of moisture availability and temperature can influence the
formation and evolution of clouds and precipitation. Studies have demonstrated that smoke
perturbations may drive widespread changes to these properties (e.g., Yu et al., 2007; Zhang et al.,
2008b; Lee et al., 2014; Herbert et al., 2021), thus influencing the overall response of the cloud field.
AIRS observations of total column water vapour ($QV_{column}$) and relative humidity at the surface
($RH_{surface}$), together with ERA5 reanalysis data of the 2-metre temperature ($T_{2m}$) are collocated with
AOD in **Figure 9**. The observations suggest that the moisture content of the column and boundary layer

(BL) generally decrease as AOD increases, though there is an initial increase at low AOD values. $QV_{column}$ will be primarily influenced by local changes in precipitation and surface fluxes that modify the water content of the BL. As observed in **Figure 3** and **Figure 4** there is an increase in convection and precipitation until AOD = 0.4, followed by a decrease. This relationship correlates well with the $QV_{column}$ sensitivity to AOD in **Figure 9**a, though there is more evident suppression of $QV_{column}$ than $P_{accum}$. This may be caused by surface cooling (due to the smoke) reducing surface fluxes of moisture (Zhang et al., 2008b), thus enhancing the drying of the BL. The sensitivity of $RH_{surface}$ is strongly influenced by $QV_{column}$ and displays a similar relationship though suppression at higher values of AOD is more pronounced, possibly due to an overall warming of the BL. $T_{2m}$ from ERA5 increases as a function of the AOD; smoke strongly absorbs solar radiation and results in anomalous heating which may explain the increase, however, $T_{2m}$ only increases by ~0.5 K over the whole range of AODs which suggests other sources influence the temperature. Surface cooling due to the overlying smoke will reduce the surface sensible heat flux, which may counteract some of the heating. The collocated data show that the large-scale environment changes alongside the AOD, and is likely driven by changes to the convective activity over the region and changes to surface fluxes due to ARI processes. The influence of large-scale external drivers to these conclusions are discussed in Section 3.6.

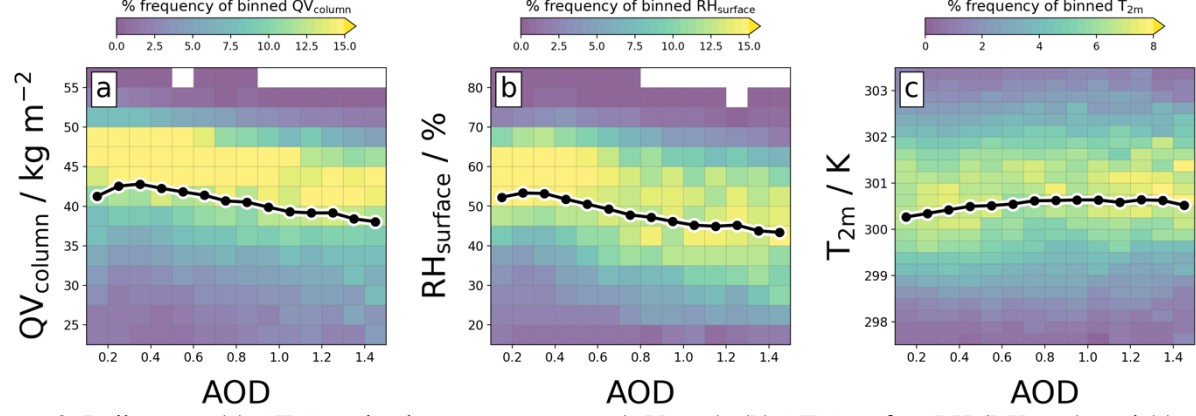

**Figure 9**. Daily mean (**a**) AIRS total column water vapour ($QV_{column}$), (**b**) AIRS surface RH ($RH_{surface}$), and (c) ERA5 2-metre temperature ($T_{2m}$) as a function of AOD. Joint-histograms show % frequency of each variable binned by AOD, and black lines show the mean in each AOD bin.

## 3.5 Top-of-atmosphere radiative effects

Collocated all-sky CERES retrievals from both TERRA and AQUA overpasses provide us with the TOA radiative impact of smoke and a means to corroborate previous findings from the MODIS retrievals.

The $SW_{TOA}$ flux is largely determined by the underlying albedo, so is function of the cloud fraction, cloud optical thickness, AOD, and surface albedo. **Figure 10** shows increases in AOD are correlated with a decrease in net downwards $SW_{TOA}$ (cooling) with a maximum cooling effect of 50 $Wm^{-2}$ in both time periods. In clear-sky conditions extinction from smoke aerosol results in less solar radiation at the surface and increases outgoing SW radiation. A 1D radiative transfer model (ecRad; Hogan and Bozzo, 2018) was used to estimate the TOA SW radiative effects due to smoke in the presence of clouds over the Amazon; output is shown in Figure S5 in the supplementary material. For a typical smoke $SSA_{550nm}$ of 0.92 over the region (Palácios et al., 2020; Rosário et al., 2011) an AOD perturbation of 1.5 results in a $SW_{TOA}$ clear-sky instantaneous aerosol radiative effect on the order of -40 $Wm^{-2}$, but is strongly offset towards positive values when even small cloud coverage is present (see Figure S5). Therefore, although the presence of smoke aerosol is potentially contributing towards the negative correlation in $SW_{TOA}$ the changes to cloud properties are likely the primary driver of the observed relationship.

540

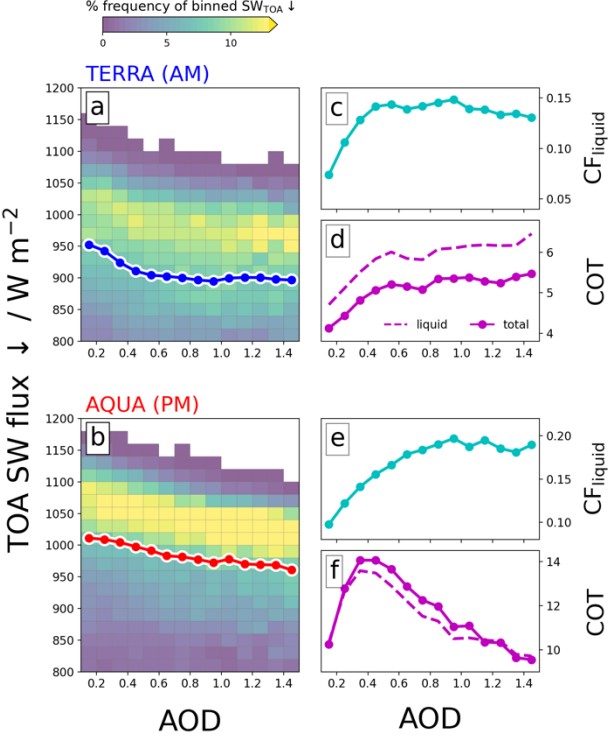

541
**Figure 10**. CERES TOA net downward SW flux as a function of AOD (**a – b**) for the morning TERRA overpass
(**a**) and afternoon AQUA overpass (**b**). Joint-histograms show % frequency of $SW_{TOA}$ binned by AOD, and
coloured lines show the mean in each AOD bin. Column on the right (**c – f**) shows MODIS retrieved mean $CF_{liquid}$
(**c, e**) and mean COT (**d, f**) binned by AOD for the corresponding satellite overpasses. COT is shown for both
$COT_{total}$ (solid) and $COT_{liquid}$ (dashed).

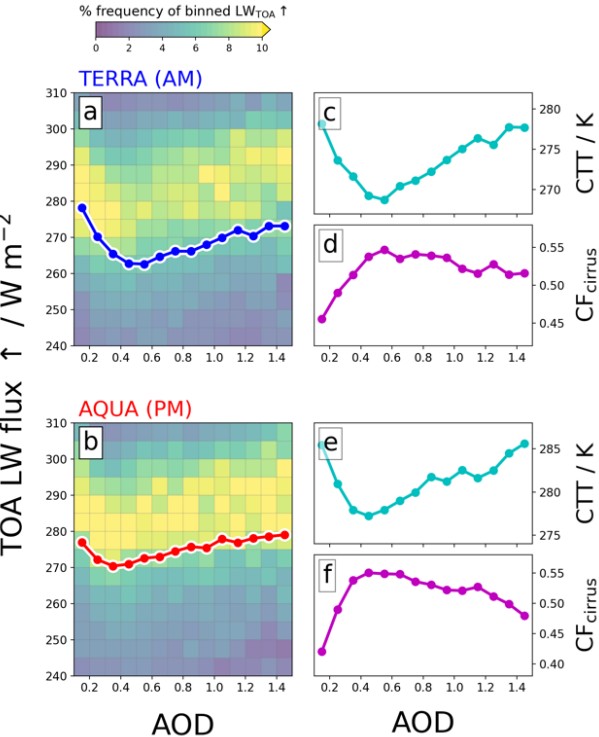

547
**Figure 11**. CERES outgoing TOA longwave flux as a function of AOD (**a – b**) for the morning TERRA overpass
(**a**) and afternoon AQUA overpass (**b**); joint-histograms show % frequency of $LW_{TOA}$ binned by AOD, and
coloured lines show the mean in each AOD bin. Column on the right (**c – f**) shows MODIS retrieved mean cloud
top temperature (**c, e**) and cirrus fraction (**d, f**) binned by AOD for the corresponding satellite overpasses.

The morning relationship is largely driven by the change in $CF_{liquid}$ across the domain (**Figure 10**c)
which increases by +100% at AOD = 0.5 then slowly decreases. COT (**Figure 10**d) gradually increases
with AOD, and is largely controlled by the change in $CF_{liquid}$ and the increase in cloud LWP (**Figure 2**).
The afternoon overpass shows a consistent increase in the outgoing $SW_{TOA}$ with AOD but of smaller
magnitude than in the morning. The relationship is well correlated with the MODIS-retrieved $CF_{liquid}$
and COT that have contrasting trends (**Figure 10**e and f); $CF_{liquid}$ consistently increases with AOD,
whereas for AOD > 0.4 COT counteracts these changes, resulting in a weakly decreasing $SW_{TOA}$
correlation.
These results provide evidence that the $SW_{TOA}$ radiative effect from smoke is strongly influenced by
the widespread changes to the cloud regimes in the region via ACI and ARI rapid adjustments. The 2d-
histograms in **Figure 10** show considerably more variability in the TERRA overpass than in the AQUA
overpass, suggesting that the background-state of the cloud field and environment in the morning plays
an important role in how it responds to the smoke. Conversely, the afternoon is more centered around
the impact to the convection. The cooling trend also suggests that the smoke is not predominantly
elevated above the cloud field. If this were the case we would expect a reduction in scene albedo as
AOD increased, driving a warming trend accentauted by the increasing $CF_{liquid}$ trend. This supports our
assumption made in Section 2.3 that the smoke is largely confined to the BL.

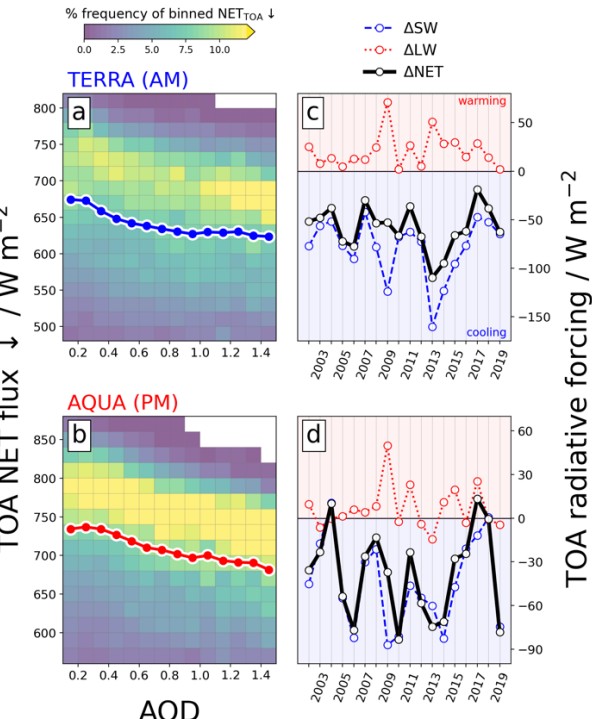

**Figure 12**. CERES TOA net incoming flux as a function of AOD (**a – b**) for the morning TERRA overpass (**a**)
and afternoon AQUA overpass (**b**); joint-histograms show % frequency of $NET_{TOA}$ binned by AOD, and coloured
lines show the mean in each AOD bin. Column on the right (**c – d**) shows the September mean TOA radiative
forcing (from 0.0 < AOD < 0.2 to 0.8 < AOD < 1.0) for SW, LW, and NET components; negative values represent
a cooling and vice versa.
Changes to the outgoing $LW_{TOA}$ flux will be driven by modification to column-integrated phases of
water, and their vertical distribution. **Figure 11** shows considerable non-linear behaviour between
$LW_{TOA}$ and AOD, apparent in both satellite overpasses. Initially mean $LW_{TOA}$ decreases with AOD until
AOD ≈ 0.4, then increases. The relationship is more apparent in the morning (**Figure 11**a) than in the
afternoon (**Figure 11**b) with reductions in $LW_{TOA}$ of -20 Wm$^{-2}$ and -10 Wm$^{-2}$, respectively. The
behaviour is well explained by the change in MODIS-retrieved CTH, which also correlate well with
$CF_{cirrus}$. The $CF_{cirrus}$-AOD relationship in both overpasses (**Figure 11**d and f) is a result of both modified
convective activity in the afternoon (**Figure 3**) and changes to the cloud-top $RE_{ice}$ (**Figure 7**c and d).
At AOD < 0.4 convection is enhanced and smaller ice particles drive extensive long-lived cirrus clouds,
whilst at higher AOD convection is suppressed (or less frequent) and ice particle sizes tend to be larger,
resulting in lower cirrus coverage. The different magnitudes in $LW_{TOA}$ appear to be a feature of the
diurnal cycle, with colder mean CTT in the morning than in the afternoon (**Figure 11**c and e) driving a
stronger sensitivity to AOD for the same decrease in CTT. An interesting feature occurs at high AOD:
the mean CTT in both overpasses is the same maximum value at very low and very high AOD, yet
$CF_{cirrus}$ does not return to the same coverage. This may be a result of ARI processes heating the smoke
layer and environment, increasing the CTT. The $LW_{TOA}$ results demonstrate that convective activity in
the afternoon (and modifications as result of smoke) produces long-lived cirrus anvil clouds that persist
throughout the night, driving the observed $LW_{TOA}$-AOD relationship in the morning.
$NET_{TOA}$ fluxes and their relationship with AOD are shown in **Figure 12**. The nonlinearity of $SW_{TOA}$
and $LW_{TOA}$ largely counteract each other, resulting in a consistent and largely linear negative
relationship between $NET_{TOA}$ and AOD (**Figure 12**a and b). In both overpasses the mean NET flux
reduces by ~50 $Wm^{-2}$ when AOD = 1, which represents a considerable aerosol radiative forcing and
pronounced cooling effect in this region. The interannual variability of SW, LW and NET components
of the radiative forcing (RF) calculated between 0.0 < AOD < 0.2 and 0.8 < AOD < 1.0 are shown in
**Figure 12**c and d. $RF_{NET}$ is consistently negative in the AM and largely negative in the PM throughout
the timeseries, though there is clearly some interannual variability in the latter timeframe. The
components $RF_{SW}$ and $RF_{LW}$ oppose each other, with the LW warming from enhanced anvil coverage
acting to partially counteract the SW cooling from changes to the liquid cloud coverage and optical
thickness, though $RF_{SW}$ dominates the $RF_{NET}$ magnitude and variability. The RF components suggest
that changes to anvil properties (LW) plays a minor role, yet a comparison with **Figure 10** and **Figure
11** show that below AOD = 0.4 the cooling is primarily driven by the changes in liquid cloud (SW),
whereas for higher loadings of AOD the reduction in anvil coverage (LW) has a more pronounced role
in driving the relationship. Additionally, the LW warming will dominate the radiative effect during the
night, and may play a more important role in the full diurnal cycle, though likely not to the extent as
estimated for deep convection over tropical oceans (Koren et al., 2010a).

## 3.6 Internalised response vs external drivers

An important question to ask is whether the sensitivity of the environment to AOD presented here is
a result of an internal response of the atmosphere over the Amazon rainforest, or an artefact of large-
scale driving meteorological conditions. These conditions may be seasonal-scale perturbations to the
transport of temperature and moisture to the region, or shifts in the climatological mean wind direction,
that result in drought susceptible conditions that may be more favourable for high AOD. In this event
the sensitivity of AOD and the widespread transition of cloud regimes that we have presented here may
be flawed.
Climatologically, the Amazon rainforest in September is characterised by easterlies that supply the
region with moisture from the Atlantic Ocean (see Figure S6a). Southerly winds originating from over
the continent may result in anonymously dry air, driving anomalous meteorological conditions and high
AOD. **Figure 13**a and b show the AOD-binned $NET_{TOA}$ subset by ERA5 collocated wind direction,
ranging from northeasterlies (200 º N) to southeasterlies (300 º N). The subsetted data show that the
cooling trend in $NET_{TOA}$-AOD is present for all wind directions, though there is variation in the
magnitude, most notably in the AQUA afternoon overpass where winds other than easterlies result in a
weaker cooling effect. A histogram of $QV_{column}$ as a function of wind direction (Figure S7) shows that
northerly and southerly winds exhibit lower loadings of water than easterlies. As the cooling trend in
the afternoon is driven by changes in convection, it is likely that the drier air masses tend to produce
weaker background convective activity as CAPE is reduced, and therefore weaken the sensitivity of the
environment to AOD perturbations. This result is similarly observed when the $NET_{TOA}$ is subset by
AIRS $QV_{column}$ in **Figure 13**c and d: the cooling trend persists but is weaker for drier airmasses,
especially for $QV_{column} < 35$ kg m$^{-2}$.

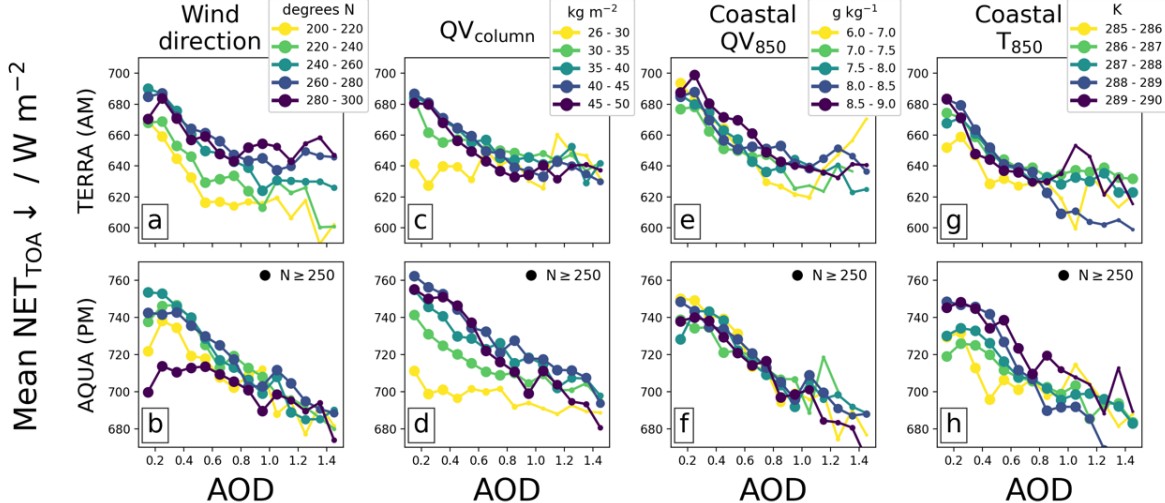

**Figure 13**. Mean $NET_{TOA}$ as a function of AOD subset by wind direction from due north (**a – b**) where 270°
describes an easterly, total column water vapour (**c – d**), mean water vapour content of the coastal boundary layer
(**e – f**), and mean temperature of the coastal boundary layer (**g – h**). Top row is for the TERRA overpass and
bottom row for the AQUA overpass. The size of each circle gives a representation of how many scenes are
included in the mean, with a maximum size shown for N ≥ 250.

Collocated meteorological variables (e.g., moisture, winds) may be influenced by the presence of
aerosol, weakening the robustness of the analysis. To account for this, we can subset the data for the
large-scale meteorology influencing the region. Data is constrained for climatological easterlies that
constitute 50% of the most frequent wind directions (Figure S6a), giving us some confidence that air
advected into the region comes from the Atlantic coast. A region due east of the analysis domain off
the coast (Figure S6b) is used to determine mean meteorological properties from ERA5, including
temperature and water vapour content at 850 hPa ($T_{850}$ and $QV_{850}$); using coastal values removes any
influence from the land-surface and associated processes. Back-trajectory analysis using the Hybrid
Single-Particle Lagrangian Integrated Trajectory (HYSPLIT) model (Stein et al., 2015) shows that air
parcels in the analysis domain tend to originate in the boundary layer in the coastal region (hence 850
hPa), taking ~ 5 or more days to reach the domain. We therefore temporally collocate the constrained
satellite dataset with mean $T_{850}$ and $QV_{850}$ from the coastal domain with an offset of -5 days. **Figure 13**
(e – h) shows the AOD-binned mean $NET_{TOA}$ subset by $QV_{850}$ and $T_{850}$ at the coast. The data, spanning
3 g kg$^{-1}$ and 5 K, shows a consistent cooling trend with almost no variation from $QV_{850}$ and slightly
weaker cooling for lower $T_{850}$ (cooler advected air may reduce CAPE); this analysis supports the
previous results.

In the final analysis, we look at the influence of climate-scale circulation anomalies/patterns such as
ENSO, PDO and the AMO. El Nino Southern Oscillation (ENSO) is largely a phenomenon that impacts
the Pacific Ocean though there have been links made between drought conditions in the Amazon and
positive phases of the ENSO (Jimenez et al., 2021; Jiménez-Muñoz et al., 2016; Aragão et al., 2018).
Similarly, the Pacific Decadal Oscillation (PDO) has also been linked to influencing the Amazon dry
season (Aragão et al., 2018). The Atlantic Multidecadal Oscillation (AMO) impacts tropical Atlantic
sea surface temperatures and the position of the Intertropical Convergence Zone, which can drive

drought conditions over the Amazon (Boulton et al., 2022; Ciemer et al., 2020; Yoon and Zeng, 2010). If strong correlations between these phenomena and the AOD over the domain are evident, this would mean it is difficult to separate the two. Conversely, if there is no clear evidence of the phenomena driving the AOD variability then this would suggest that changes to the cloud field and environment (with respect to AOD) are more heavily influenced by local perturbations – i.e., the smoke. The same applies with other variables such as the $QV_{column}$ or $RH_{surface}$. **Figure 14** shows the September mean AOD, $QV_{column}$, and $NET_{TOA}$ each year of the timeseries as a function of the corresponding ENSO, PDO and AMO indices (averaged over August and September). There are no strong correlations evident for any pairings, and the strongest correlations are at odds with what we would expect. For example, positive ENSO years are generally associated with drought conditions, yet we observe lower AOD and lower $QV_{column}$. Although the sample size is small, this does suggest that the AOD is driven by localised processes, such as anthropogenic sources, rather than large-scale circulation anomalies. Similarly, the $QV_{column}$ is not significantly influenced by these phenomena, and possibly primarily driven by local sources of moisture. Although this is not an extensive nor entirely quantitative analysis we would expect there to be more correlation if these large-scale circulation anomalies were driving the strong responses that are evident from the MODIS, AIRS and CERES collocated retrievals. AIRS collocated data (**Figure 9**) show that low AOD scenes are typically more moist than high-AOD scenes. $RH_{surface}$ decreases more rapidly with AOD than $QV_{column}$ which suggests temperature is also increasing at high AOD, this would be consistent with a localised heating of the smoke layer.

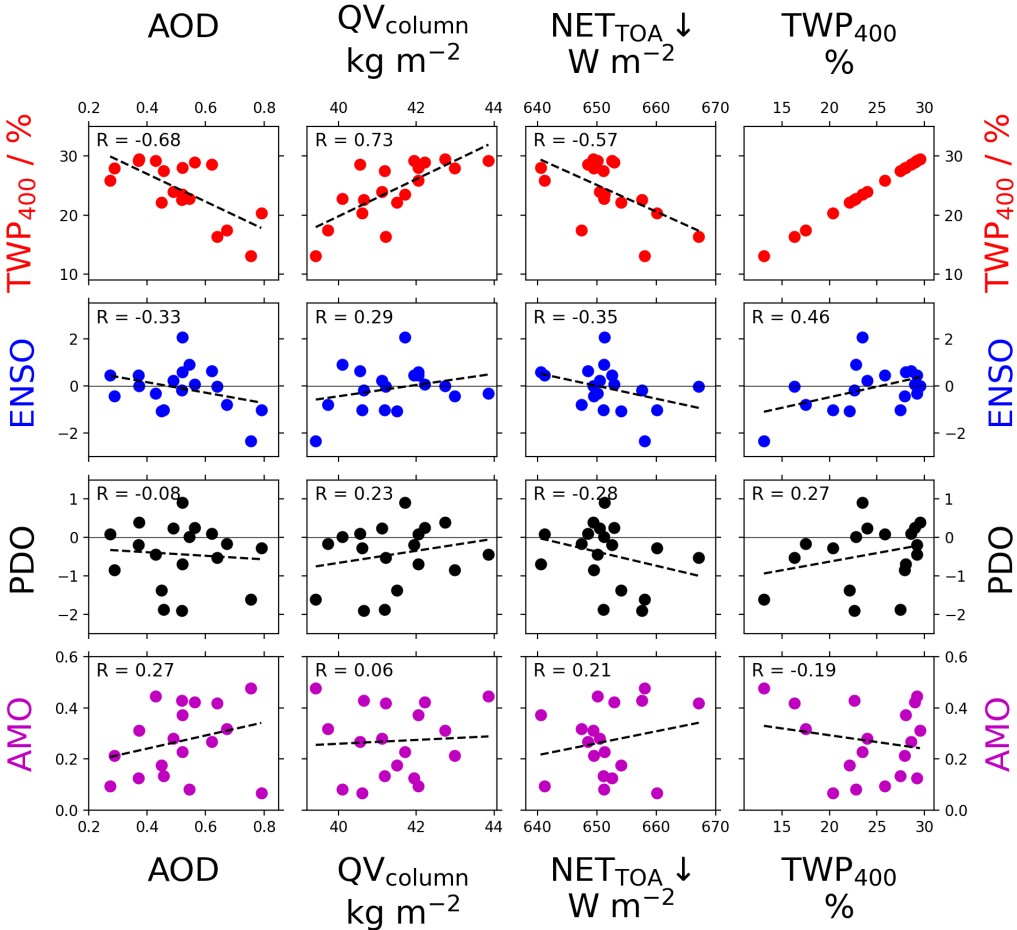

**Figure 14**. September mean AOD (left column), $QV_{column}$ (center column), and $NET_{TOA}$ (right column) from the domain for each year as a function of the corresponding (from the top row downwards) percentage of domain where TWP > 400 gm$^{-2}$, August-September mean ENSO index, PDO index, and AMO index. The dotted line in each plot shows the linear regression between the two datasets, with the corresponding R value shown at the top of each plot. $TWP_{400}$ is used as a metric to describe the 'convective nature' of the domain.

**Figure 14** also shows the September mean percentage of domain where TWP > 400 gm$^{-2}$ (named
TWP$_{400}$ in the plot) used to indicate the 'convective nature' of the season; higher values will be
associated with more numerous deep clouds throughout the domain hence a reasonable proxy for more
convection. Here we see strong relationships between AOD and convection, as well as the QV$_{column}$ and
NET$_{TOA}$. The regressions suggest that, on seasonal timescales, high-AOD years coincide with
suppressed convection, a drier atmosphere, and a net cooling radiative effect.

## 4. Discussion and Conclusions

In this study we used spatially and temporally collocated observations and estimates from multiple
satellite instruments and datasets to examine smoke-cloud-radiation interactions over the Amazon
rainforest during the month of September. We found evidence that smoke drives widespread changes
to the cloud field over the region consistent with ACI and ARI processes. **Figure 15** shows a schematic
summarising the main findings.

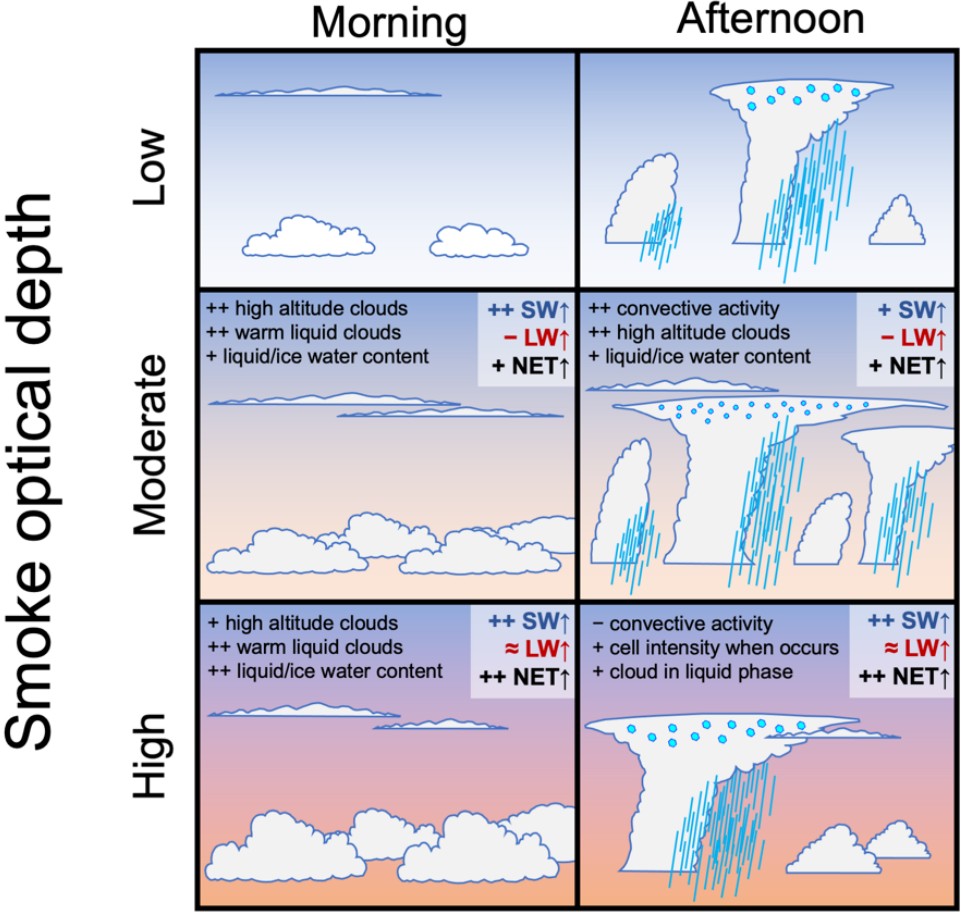

**Figure 15**. Summary of results from an 18-year timeseries of collocated MODIS, CERES, and AIRS observations
onboard TERRA (morning overpass) and AQUA (afternoon overpass) satellite platforms, combined with IMERG
precipitation estimates, during the peak biomass burning month of September over the Amazon rainforest. Panels
are shown for low (AOD < 0.1), moderate (AOD = 0.4), and high (AOD > 1.0) smoke optical depths. Annotations
are included to highlight primary responses to the cloud field and TOA outgoing radiation fluxes as compared to
the background low AOD scene; symbols depict increases (+), substantial increases (++), decreases (−), and
relatively little change (≈).
The Amazon atmosphere is very sensitive to low-to-moderate loadings of smoke where AOD ≤ 0.4.
In the morning the smoke perturbation coincides with increases in the warm phase cloud coverage and
cloud optical thickness, consistent with a suppression of the warm rain process, whilst in the afternoon

there is a considerable enhancement in the formation and development of deep convection, enhancing daily accumulated precipitation and intensity, and high-altitude cloud coverage. The high-altitude clouds persist throughout the night and into the morning, possibly enhanced by smaller ice particle sizes. The increased coverage and optical thickness of liquid clouds enhances the scene albedo, resulting in a negative SW forcing; enhanced cirrus coverage partially offsets this via a decrease in LW, resulting in a negative TOA net forcing.

At higher loadings of smoke where AOD > 0.4 the liquid cloud coverage in the morning remains relatively stable with small increases in the cloud optical depth resulting in enhanced TOA cooling; an overall drying and warming of the boundary layer may play a role in limiting the cloud coverage extent. A primary response of the atmosphere in the afternoon is an overall suppression of convection, consistent with a stabilization of the atmosphere via surface cooling and elevated heating by ARI processes (Herbert et al., 2021). A reduction in cumulative precipitation and cirrus cloud coverage is consistent with the suppressed convection, along with a shift from the ice phase to liquid phase as mean cloud vertical extent decreases. At very high AOD accumulated precipitation remains comparable with background (very low AOD) scenes, despite weaker convection across the domain, which may suggest fewer, more intense, convective cells, consistent with the simulations of Herbert et al. (2021) and observations of delayed and more intense precipitation (Andreae et al., 2004; Gonçalves et al., 2015) though this would require further investigation to confirm.

These results are generally consistent with previous studies but also help to fill in some important knowledge gaps. Previous studies have focused on MODIS-AQUA retrievals to study the response of warm liquid clouds to aerosol over the region. Koren et al. (2004) used retrievals from the dry season of 2002 and reported a pronounced decrease in cloud fraction as the smoke optical depth increased. Using a similar methodology Yu et al. (2007) analysed data for two consecutive years and found opposing correlations (negative in 2002; positive in 2003). Ten Hoeve et al. (2011), focusing on a smaller domain and over four years, reported a consistent increasing cloud fraction with AOD; the authors found that the collocated CWV of the scene strongly influenced the cloud fraction and proposed that this behaviour may explain the opposing correlations in Yu et al. (2007). In our study we do not subset for one cloud type, and instead consider all clouds, making a direct comparison difficult. However, comparing to $CF_{liquid}$ in our study for the AQUA overpass, we observe a shift towards higher coverage as AOD increases (**Figure 8**b and **Figure 10**e) for all years in the timeseries (not shown), consistent with Ten Hoeve et al. (2011) but not with Koren et al. (2004) and Yu et al. (2007). The inconsistencies may be explained by the differing methodologies, in that the authors removed scenes with cloud fractions > 0.8 whereas in our study we do not. Subsetting our data to remove scenes with $CF_{liquid}$ or $CF_{total}$ > 0.8 has considerable impact on our results, as it removes a lot of data from the higher AOD scenes (**Figure 8**), biasing the dataset towards lower cloud fractions. The result of subsetting our data is a negative $CF_{liquid}$-AOD relationship at higher AOD and a weaker TOA radiative effect though of the same sign. This suggests that results from previous studies may be biased towards lower cloud fractions. However, a caveat is that the primary reason for restricting high CF values is to reduce misclassification of clouds and aerosol (Koren et al., 2010b). To test this further we used level-2 MODIS products (10 km resolution) to compare with the coarser (1 degree) level-3 data. Cloud products at 5-km resolution are regridded to 10-km resolution and spatially/temporally collocated with the 10-km aerosol product. The comparison is shown in Figure S8 of the supporting information. First, the distribution of level-2 AOD (Figure S8a) and $CF_{total}$ (Figure S8b) within each 1-degree pixel shows very good agreement between scales, with reasonable variability around the mean and median. These illustrate that the AOD (and cloud response) is widespread amongst the region, rather than focused within single plumes of smoke or single cloud features. Secondly, at high values of AOD the retrieved 10-km cloud fraction is increasingly close to 1 with less variability than at lower values. We would expect more variability across the 1-degree pixel if there was widespread misclassification occurring. We also perform the same analysis as in Section 3 to test our conclusions on this finer-scale dataset. **Figure 16** shows the same trends, and of similar magnitude, to those observed using the 1-degree data.

This analysis helps to support our conclusions and method, though we cannot rule out misclassification,
so some caution should be applied until further work can corroborate these findings.

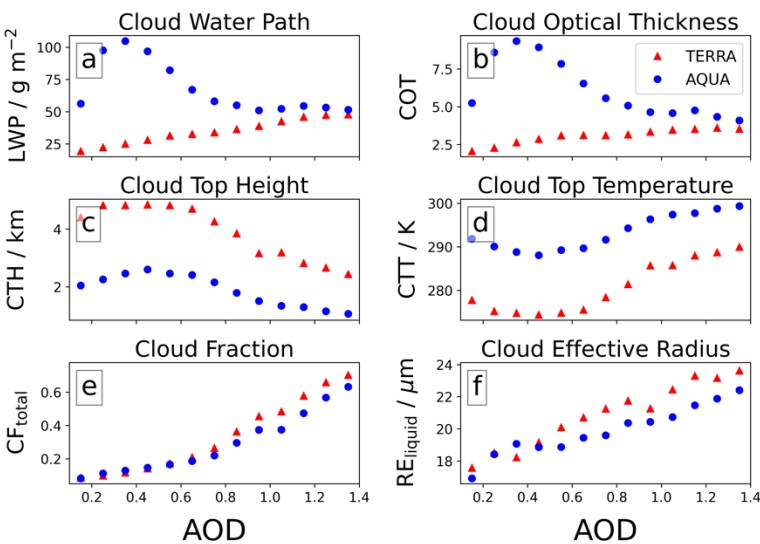

**Figure 16.** Cloud properties from the MODIS 5-km cloud product binned by MODIS 10-km AOD for the TERRA
(red triangles) and AQUA (blue circles) satellite overpasses. The mean values (symbols) are taken from all
spatially and temporally collocated grid points within the AOD bin (5-km product regridded to 10-km). Data are
for September 2002 to 2019 inside the domain 68°E to 58°E, 9°S to 1°S.
Our conclusions may additionally explain behaviour reported by Koren et al. (2008). In the study the
authors examine the relationship between low cloud fraction and AOD in MODIS-AQUA data. At
higher AOD the authors find that subsetting the data to increasingly lower cloud fractions results in an
increasingly negative $CF_{liquid}$-AOD relationship, attributed to greater sensitivity of low cloud-fraction
scenes to aerosol absorption. This behaviour was also seen in our data when we subset the data to
remove high $CF_{liquid}$ scenes, therefore the results from Koren et al. (2008) could be alternatively
interpreted as a result of dampening the underlying pathway, which is a pronounced shift from low to
high $CF_{liquid}$ scenes as AOD increases and modifies the widespread convective nature of the region. It
is also possible that both processes are simultaneously occurring and contributing to the overall response
of the cloud field.
A key process influencing the diurnal cycle of cloud cover and vertical distribution is via the
modification to convection in the afternoon, driven by ARI at high AOD and ACI and/or thermal
buoyancy at low AOD. We observe increasingly suppressed convection and precipitation for AOD >
0.4 during the AQUA overpass; this is consistent with modelling studies that report ARI-driven
stabilization of the lower atmosphere (Herbert et al., 2021; Liu et al., 2020; Martins et al., 2009; Wu et
al., 2011) and suppressed (or delayed) convection with similar impacts to precipitation. Field studies
from the region have similarly reported suppressed or delayed peak precipitation rates (Andreae et al.,
2004; Bevan et al., 2008; Camponogara et al., 2014; Gonçalves et al., 2015), and remote observations
from Koren et al. (2008) show a tendency for shallower convective clouds (less vertical extent) under
high aerosol loading. The invigoration of convection at AOD < 0.4 in our observations suggests an
important process that has considerable implications for the region. Koren et al. (2008) report an
increase in cloud fraction and taller convective clouds at small AOD perturbations, and Ten Hoeve et
al. (2011) reported similar behaviour with $COT_{liquid}$. This is consistent with ACI-induced warm phase
invigoration in shallow convection and in the warm base of deep convective cells (Marinescu et al.,
2021; Koren et al., 2014; Seiki and Nakajima, 2014; Igel and van den Heever, 2021; Dagan et al., 2020),
or through anomalous thermal buoyancy due to the fire itself (Zhang et al., 2019). The reduction in
cloud top $RE_{ice}$ (**Figure 7**c and d) with AOD for high-IWP scenes suggests more cloud droplets are
reaching the freezing level; this may due to ACI processes or enhanced aerosol activation through
thermally-induced anomalous buoyancy, making attribution of the dominant mechanism difficult.
The analysis suggests an important inflection point in the Amazonian atmosphere's response to
aerosol at AOD ≈ 0.4. This value represents close to 50% of the retrieved AOD values over the time
period analysed (**Figure 1**c), suggesting that in the near-present climate enhanced convection is as likely
as supressed convection. Current trends and future projections suggest biomass burning frequency and
scale will increase throughout the Amazon rainforest (Stocker et al., 2013; Boisier et al., 2015); this
will increase the likelihood of deep convection being supressed and overall result in reduced cumulative
precipitation to the region and potentially act as a positive feedback to fire activity and AOD.
Simultaneously, increases in AOD are correlated with an overall brightening of the scene albedo
(**Figure 10**) and a warmer, drier, boundary layer (**Figure 9**). Together with reduced precipitation there
may be important impacts to the Amazonian biosphere and ecosystem.
The pronounced diurnal cycle in the response of the clouds to aerosol is consistent with high-
resolution modelling studies from the Amazon (Herbert et al., 2021) and over Borneo (Hodzic and
Duvel, 2018), a region similarly dominated by biomass burning aerosol. The same contrasting responses
in LWP and IWP were found when analysing scenes independently (**Figure 2** and **Figure 7**) and the
domain as a whole (Figure S2), suggesting the signal is independent of scale. These strong repeatable
signals point towards the possibility of using the amplitude of the diurnal cycle in key cloud properties
as an important source of information for constraining global ARI and ACI effects on the climate. This
could be applied to both earth-system models and observations, and work towards reducing the
uncertainty in current forcing estimates (Forster et al., 2021), with the caveat that current earth-system
models used to produce the forcing estimates do not fully capture these convective processes. This study
highlights the need for explicit treatment of convection in climate models.
Both overpasses suggest AOD drives an overall SW cooling at the TOA due to changes in cloud
properties. This is at odds with the theoretical model proposed by (Koren et al., 2004) who estimated
that cloud field adjustments due to smoke (cloud thinning) over the Amazon would counteract some of
the cooling, which suggests that the widespread radiative impact of smoke aerosol over the Amazon
rainforest is more important than previously thought. We also find important changes to high-altitude
cloud coverage, likely from deep convective outflow, which impact the outgoing LW at the TOA.
Unlike over the tropical oceans (Koren et al., 2010a), these are of secondary importance when compared
to changes in SW, but will influence the daily mean radiative effect due to their dominating role during
the night. This study would benefit from using geostationary satellite data from GOES to validate our
findings and extend the analysis throughout the full diurnal cycle, but would require well validated
aerosol retrievals which are currently unavailable.

## Author Contribution

RH designed the study and acquired the datasets. RH wrote the necessary scripts and analysed the
dataset. RH prepared the manuscript with contributions from PS.

## Competing Interests

Some authors are members of the editorial board of journal ACP. The peer-review process was
guided by an independent editor, and the authors have also no other competing interests to declare.

## Acknowledgements

This research was supported by the European Research Council (ERC) project constRaining the
EffeCts of Aerosols on Precipitation (RECAP) under the European Union's Horizon 2020 research and
innovation program with grant agreement no. 724602 and from the European Union's Horizon 2020
research and innovation program project Constrained aerosol forcing for improved climate projections
(FORCeS) under grant agreement No 821205.

## Data Availability

All satellite datasets used in this analysis are available online. MODIS datasets are available via the
NASA Level-1 and Atmosphere Archive & Distribution System (LAADS) Distributed Active Archive
Center (DAAC) at https://ladsweb.modaps.eosdis.nasa.gov/archive/allData/61/. IMERG daily and
instantaneous data are available via the NASA Goddard Earth Sciences Data and Information Services
Center (GESDISC) at https://gpm1.gesdisc.eosdis.nasa.gov/data/. ERA5 reanalysis datasets from the
European Centre for Medium-Range Weather Forecasts (ECMWF) are available via the Natural
Environment Research Council (NERC) Centre for Environmental Data analysis (CEDA), accessed via
https://data.ceda.ac.uk/badc/ecmwf-era5/. AIRS data is available via NASA's Earth Science Data
Systems (ESDS) program at https://www.earthdata.nasa.gov/. AERONET data is available from
https://aeronet.gsfc.nasa.gov/. CERES datasets are available at https://ceres.larc.nasa.gov/. HYSPLIT
back trajectories were performed online at https://www.ready.noaa.gov/HYSPLIT.php. The ecRad
offline radiative transfer model is available via github at https://github.com/ecmwf-ifs/ecrad. This work
used the ARCHER2 UK National Supercomputing Service (https://www.archer2.ac.uk). The spatially
and temporally collocated datasets (at one- and two-degree resolution) are available alongside the
relevant scripts for reproducing all figures at http://doi.org/10.5281/zenodo.7007220.

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
