# Peer review of "Satellite Observations of Smoke-Cloud Radiation Interactions Over the Amazon Rainforest"

_Atmospheric Chemistry and Physics, 2022_

## Author Comment (AC1)

**Response to reviewers of ACPD manuscript "Satellite Observations of Smoke-Cloud-Radiation Interactions Over the Amazon Rainforest" by Ross Herbert and Philip Stier.**

Firstly, we would like to thank the two reviewers for their valuable comments and suggestions. Below we go through each comment (indented) and provide a point-by-point response. Revised sections/paragraphs of text in the revised manuscript are italicised.

**Review # 1**

> A review of "Satellite Observations of Smoke-Cloud-Radiation Interactions Over the Amazon Rainforest" by Herbert and Stier
>
> The authors use a suite of satellite and reanalysis measurements to explore the effect of smoke on clouds and the overall radiative forcing. They do it for the dry season over the Amazon for 18 years of data. They describe part of the diurnal cycle by comparing associations between smoke AOD and cloud properties during TERRA and AQUA passing times.
>
> The methods are well described, and the analysis for such a time series shows convincing results. However, the interpretation of the results is not clear enough, and there are a few important points that the author should explore more:

Comment:

> These is an inherent problem in the cloud detection results in cases of high AOD. The higher the AOD, the more likely that smoke areas will be classified as clouds. Such a problem dictates a positive bias of the reported cloud fraction with AOD. Moreover, an average AOD of 0.8 in one-degree resolution might result from much higher AOD pixels in the core retrieval resolution. I encourage the authors to explore days with high AOD and to look at both aerosol and cloud properties in a few level 2 data to understand the inputs to the level 3 averages. It is easy to see the above problem. It influences their reported results on AOD-cloud associations in the high AOD regimes.

Response:

We thank the author for this comment. We also saw this as an important source of potential bias and endeavoured to explore this in the manuscript. However, we agree with the reviewer that this requires more attention and have therefore performed additional analysis using the level-2 10km aerosol product, and the level-2 5km cloud product as suggested. This comment follows similar arguments to the one below, therefore we combine the response below.

Comment:

> On the same note – Overland in cloudy conditions, the AOD retrievals could be sparse. Therefore many level 3, one-degree pixels could report an average AOD that originates from only a few true retrieved pixels. In particular for cases with high cloud fractions. Again creating a bias in the data reliability as a function of CF. The author should explore it and avoid using level 3 pixels that are the outcome of aerosol retrievals from a small fraction of the domain.

Response:

We agree with the reviewer and have provided additional analysis to explore this and reinforce our methods and conclusions.

We used the level 2 MODIS 10-km aerosol products (MYD04_L02 and MOD04_L02) in combination with the MODIS 5-km cloud products (MOD06_L2 and MYD06_L2) to explore the finer resolution products. The cloud product was regridded to a 10-km grid and spatially/temporally collocated with the 10-km aerosol product. Each 1-degree grid box used in the manuscript analysis contains roughly 121 of these finer resolution gridpoints. Data was obtained for the same analysis period (September 2002 to 2019) on an overlapping domain (68°E to 58°E, 9°S to 1°S) focused on the region with highest AOD observations. We use the combined Dark Target + Deep Blue product giving AOD at 550 nm.

We performed a similar analysis as with the 1-degree dataset, binning the cloud product data with the aerosol AOD for both TERRA (morning) and AQUA (afternoon) overpasses. Figure R1 shows the arithmetic mean within each AOD bin. For both overpasses we see consistent trends (with AOD) as with the 1-degree analysis, specifically an increase in LWP and CTH up to an AOD of 0.4, followed by a gradual decrease for the afternoon, and an overall increase in LWP with AOD for the morning overpass, with strong CTH and CTT similarities between the two overpass times. Importantly, the CF also shows strong consistency with the 1-degree dataset, that being an increasing CF with AOD. This helps strengthen our assumptions that we are not seeing cloud/aerosol retrieval bias.

[Figure]

*Figure R1. Various cloud products from the MODIS 5-km cloud product binned by MODIS 10-km AOD (combined DB+DT) for the TERRA (red triangles) and AQUA (blue circles) satellite overpasses. The mean values (symbols) are taken from all spatially and temporally collocated grid points within the AOD bin (5-km product regridded to 10-km). Data are for September 2002 to 2019 inside the domain (68E to 58E, 9S to 1S).*

As suggested by the reviewer we have also explored the distribution of CF and AOD values that comprise the 1-degree dataset. For this we have obtained all 10-km gridpoints that overlap with each 1-degree grid point (for each day and overpass). The spread of the finer resolution data, compared to the coarser, provides us with information on how variable, and representative, the data are. Figure R2 below shows that there is very good agreement for both AOD and CF. AOD shows very little variability, suggesting that the AOD that is measured in the region characterised by widespread loadings of aerosol, rather than individual plumes. CF shows more variability than AOD, but still shows that the 1-degree CF data are consistent with what is observed at the finer scale. Both comparisons suggest the response of this region's atmosphere to smoke are occurring over widespread regions, rather than individual localised sources. This backs up our conclusions and hypothesis that these impacts are driving widespread changes to the environment and cloud-field as a whole, rather than individual clouds.

[Figure]

*Figure R2. Comparing total cloud fraction ($CF_{total}$) and aerosol optical depth (AOD DT+DB) from MODIS level 2 products (5-km for $CF_{total}$; 10-km for AOD) and level 3 products (1 degree resolution). Fine resolution data is binned by the coarse resolution data and displayed as a boxplot (red bar shows median; red cross shows mean). The comparisons are shown for (**a**) AOD, (**b**) $CF_{total}$, and (**c**) $CF_{total}$ at 10-km and AOD at 1 degree. Top row is for the TERRA overpass, and the bottom row for the AQUA overpass.*

Comparing the distribution of CF-10km with the AOD-1degree data can give us additional information as to whether the smoke is being misidentified as cloud. Though there is a lot of variability within each AOD bin there is a consistent trend for the distribution of CF-10km to be centered around higher CF values as AOD increases. In fact, at the higher values of AOD (>1.2) the variability becomes smaller and smaller. We may expect the opposite of this to be occurring if smoke was being misidentified as cloud, which would become increasingly apparent at higher values of CF.

We include Figure R1 into the manuscript and include Figures R2 in the supporting information. We have included references to these figures in the revised manuscript in Section 4:

*To test this further we used level-2 MODIS products (10 km resolution) to compare with the coarser (1 degree) level-3 data. Cloud products at 5-km resolution are regridded to 10-km resolution and spatially/temporally collocated with the 10-km aerosol product. The comparison is shown in Figure S8 of the supporting information. First, the distribution of level-2 AOD (Figure S8a) and $CF_{total}$ (Figure S8b) within each 1-degree pixel shows very good agreement between scales, with reasonable variability around the mean and median. These*

*illustrate that the AOD (and cloud response) is widespread amongst the region, rather than focused within single plumes of smoke or single cloud features. Secondly, at high values of AOD the retrieved 10-km cloud fraction is increasingly close to 1 with less variability than at lower values. We would expect more variability across the 1-degree pixel if there was widespread misclassification occurring. We also perform the same analysis as in Section 3 to test our conclusions on this finer-scale dataset. **Error! Reference source not found.** shows the same trends, and of similar magnitude, to those observed using the 1-degree data. This analysis helps to support our conclusions and method, though we cannot rule out misclassification, so some caution should be applied until further work can corroborate these findings.*
* * *
Comment:

> Another related front that is not covered here is related to problems of measuring aerosols in the vicinity of clouds. The authors should describe the potential problems (both in the aerosol and cloud domains) and explain why they trust the reported results.

Response:

We have added the following text to the manuscript in Section 2.3:

*A second source of potential bias may arise from the retrieval of AOD in cloudy conditions. The presence of aerosols in the vicinity of clouds can impact the retrieval of both properties: enhanced humidity close to clouds can cause aerosols to swell elevating the AOD retrievals, whilst aerosols embedded within, or below, clouds may be misidentified as cloud, thereby modifying the retrieved cloud optical properties. Finally, very high loadings of aerosols may be misidentified as cloud. These are well-known sources of retrieval bias and as such cloud masking algorithms are continually refined to separate the influence of the two. The MODIS cloud mask product in the collection 6 variants, used in this study, is constructed using 1km scale pixels and employs multi-spectral tests to identify heavy aerosol loading. Aerosol retrievals are made in clear-sky pixels, with collection 6.1 using the Dark-Target and Deep-Blue aerosol retrieval algorithms, designed to take into account the underlying surface properties. These well maintained, and extensively evaluated products (e.g., Wei et al., 2019; Huang et al., 2019; Zhang et al., 2022; Levy et al., 2013; Platnick et al., 2017) provide a robust dataset of collocated aerosol and cloud properties but may not remove all bias. Therefore, to support our analysis we will pay particular attention to aerosol-cloud misclassification, especially at high cloud fractions. We achieve this by first comparing the MODIS retrievals of AOD with those from two AERONET stations (below), and later in Section 4 repeat the analysis with level-2 data products, where we find the same conclusions.*
* * *
Comment:

> A significant part of the reported aerosol effect is linked to aerosol absorption. Absorption is hardly mentioned in the text (it is referred to vaguely as ARI, which in my opinion, is not clear enough). If indeed the authors think that aerosol absorption plays an important role in the reported effects, they should explain it more, and they should provide some measures of the importance of absorption. For example, they can use newer satellite measurements to explore the optical properties of the Amazonian smoke. As well as Aeronet measurements for the SSA. I am aware that some radiative effects calculations are presented in the SI, but it is not explaining the above points.

Response:

We thank the reviewer for the comment and suggestion. To quantitively show the absorption properties of the aerosol during the analysis period and in the domain, we have included a figure showing the daily mean SSA measured from two AERONET stations within the Amazon rainforest (Alta Floresta and Rio Branco). Figure R3e shows a histogram of SSA measured at 675nm for all days where data were available for the month of September between 2002 and 2019. Both stations show SSA peaking at around 0.93 which is consistent with other studies that have measured Amazon smoke SSA. This value demonstrates that the aerosol is indeed absorbing and can play an important role via aerosol-radiation interactions (ARI). Additionally, at longer wavelengths the SSA of dust increases and becomes less absorbing; the low SSA values shown here suggest that the aerosol loading is dominated by smoke.

We use Figure R3 in place of Figure 1 in the revised manuscript, and include the following text to the manuscript at the beginning of Section 2.1:

*AERONET stations at the Rio Branco and Alta Floresta sites provide information on the single scattering albedo (SSA) of the aerosol throughout the analysis period. These sites are situated at opposite ends of the analysis region and collocated with the climatologically highest regions of AOD (Figure 1a). Histograms of the daily-mean SSA from each station, given at 675 nm, are shown in Figure 1e. Both stations show $SSA_{675}$ ranging from values as low as 0.85 to 0.98, with a peak around 0.93. This consistent with in-situ local observations of smoke optical properties (Palácios et al., 2020; Rosário et al., 2011), providing good evidence that the aerosol in this analysis period and domain is strongly absorbing smoke. Note that mineral dust has a SSA closer to 1 at this wavelength (Di Biagio et al., 2019). We would therefore expect ARI mediated impacts via absorption of solar radiation to be a viable mechanism in this region.*

[Figure]

Figure R3. Revised version of Figure 1. This version now includes a histogram of single-scattering albedo (SSA) measured at the same two AERONET stations used for AOD observations (Rio Branco and Alta Floresta) for September 2002 to 2019. SSA is given at a wavelength of 675nm.

The manuscript ACP-2022-796 entitled "Satellite Observations of Smoke-Cloud-Radiation Interactions Over the Amazon Rainforest" by Herbert et al. presents a comprehensive analysis of 18-year time series from multiple satellite sensors over the Amazon basin. The study focuses particularly on the influence of (heavy) biomass burning smoke during the peak of the Amazonian burning season on a spectrum of atmospheric parameters, including cloud properties, top-of-atmosphere radiative flux, rainfall, and column water vapor content. The smoke influence is represented here by the satellite-measured aerosol optical depth (AOD). The study concludes that the cloud and radiative properties of the Amazonian atmosphere are strongly modified by the smoke. Among other observations, the extent of smoke influence defines if convective activity is either enhanced or suppressed. Further, an interesting contrast between the morning and afternoon cloud responses has been found. The results are thoroughly discussed/integrated in the context of previous studies and existing knowledge.

My overall recommendation is that the manuscript should be published after some relatively minor changes. From a scientific perspective, it clearly fits the scope of ACP. It is a thorough and interesting study, relevant for the field of Amazonian research and beyond. I am convinced that the study will become a useful resource and reference for researchers studying the smoke influence of atmospheric radiative transfer and aerosol-cloud interactions. The conclusions are important for discussions and predictions of the future development of the Amazon ecosystem, which is under increasing pressure from progressing climate and land use change. From a formal perspective, the quality of the manuscript is high - it is well-written and all arguments and aspects are presented clearly.

Below, you will find some general comments that I recommend to consider to further improve the quality of the manuscript:

Comment:

AOD is the main parameter representing the atmospheric smoke/aerosol concentration in this study. I suggest adding a targeted paragraph, which outlines and discusses the relationship between AOD and the actual aerosol optical properties as well as their ability to act as CCN. In other words: To what extent does AOD (as a simplified parameter) represent the relevant microphysical properties of the aerosol and what are the limitations of this?

Response:

We thank the review for their suggestion and agree this would be beneficial for the reader. We have added a paragraph to Section 2.3 to briefly discuss this. The text reads:

*In this study we are primarily interested in how widespread properties of the atmosphere change with AOD. We use AOD as a proxy for the availability of aerosols that can influence clouds both via ARI and ACI, thereby assuming that as AOD increases linearly, so does the number of aerosols that act as CCN and interact with radiation. For ARI this assumption is reasonable if the source and size distribution stays relatively constant as AOD increases. As the primary source of aerosol in this region is biomass burning, with AOD increasing linearly with the frequency of fires (Ten Hoeve et al., 2012), this is to first order a reasonable approximation. This can be similarly applied to the availability of CCN but the number activated is also dependent on properties of the atmosphere, namely the updraught speed. Herbert et al. (2021) used in-situ observations from field campaigns over the Amazon and found a positive albeit non-linear relationship between AOD and cloud droplet number concentration (CDNC). However, this is confounded by any changes to the distribution of vertical velocities as AOD changes. Given the inherent non-linearity and confounding factors between AOD and CDNC we can only say that AOD is a reasonable proxy for the availability of CCN.*
* * *
Comment:

> The influx of African smoke, which appears to occur in elevated layers, is discussed briefly on page 2. This particular aspect might profit from some more discussion in the introduction and/or results and discussion section. This relates to the particular questions (i) whether the assumption that the smoke is only/mostly in the PBL is justified and (ii) whether the current analysis provides any indications for the extent of influx of African smoke into the central and southern Amazon basin.

Response:

We thank the reviewer for their suggestions. Our assumption is based on a 6-year climatology of smoke layer heights retrieved using three instruments (Gonzalez-Alonso et al., 2019). This study provided evidence that in most cases (between 80 and 95%) the smoke was confined to the planetary boundary layer. Given the lack of widespread and long-term observations of these properties from other studies, we believe this provides justification to our assumption. However, following the suggestions by the reviewer we have extended the introduction section to discuss the position of the smoke layer in more detail, including the importance of the smoke layer, and more on the results of previous observationally studies. We also discuss this assumption in Section 3.5 following the presentation of CERES TOA fluxes.

The added text in the introduction:

*Previous studies (e.g., Koch and Del Genio (2010)) have shown that the position of smoke in relation to clouds can greatly impact the cloud rapid adjustments and ERF. Most significantly, when smoke is elevated above clouds it reduces the scene albedo, thereby driving a positive TOA instantaneous radiative effect. The vertical profile of aerosol is a difficult property to measure on the scales that we are interested in. Gonzalez-Alonso et al. (2019) used three remote sensing instruments over 6 years to construct a climatology of smoke heights over the Amazon. The authors found that smoke plumes during September are generally located below 1.5 km, with less than 5 % of smoke plume injection heights observed in the free troposphere. Some studies, focusing on the eastern edge of the Amazon rainforest, have reported the presence of smoke being transported from the African continent at concentrations that often compete with localised sources (Barkley et al., 2019; Holanda et al., 2020). Therefore, although we assume that the smoke in this analysis is predominantly within the BL and from*

*local sources, we caveat that this is not always the case. We discuss the validity of this assumption in Section 3.5.*

The added text in section 3.5:

*The cooling trend also suggests that the smoke is not predominantly elevated above the cloud field. If this were the case we would expect a reduction in scene albedo as AOD increased, driving a warming trend accentauted by the increasing $CF_{liquid}$ trend. This supports our assumption made in Section 2.3 that the smoke is largely confined to the BL.*
* * *
Comment:

> The manuscript summarized a broad spectrum of aspects and observations, with manifold links to previous studies. Because of the density of information and the length of the manuscript, it is hard to read in some sections. I have got the impression that a (slightly) different overall structure might make the study more accessible. The current structure separates a 'results section' and a 'discussion and conclusions section'. First, I think that the results section already comprises quite some discussion of the results already. Therefore, parts of the discussion section appear somewhat redundancy. I wonder if for this particular study a merged 'results and discussion section' might work better. Second, I think that the study needs a concise and short conclusion paragraph – separated from the longer discussion sections. This would make it easier for readers that just would like to grasp the essence and main conclusions.

Response:

We thank the reviewer for their suggestions. We designed the results section to provide a step-by-step story through which the reader can be introduced to the underlying results/conclusions first with a single variable (LWP) and then build upon this with other variables and observations to provide additional support. We then designed the discussions and conclusions section to bring together these separate sections into a single broad picture of understanding. We have looked into ways to keep this arrangement whilst addressing the reviewer's suggestion but feel that large portions would have to be rewritten to get a sensible and logical flow. Therefore, although we are very grateful for the reviewer's suggestion, we wish to keep the current arrangement.
* * *
**References**

[revised manuscript text omitted]